# Dalpiciclib partially abrogates ER signaling activation induced by pyrotinib in HER2+HR+ breast cancer

Jiawen Bu[1†], Yixiao Zhang[1,2†], Nan Niu[1†], Kewei Bi[1], Lisha Sun[1], Xinbo Qiao[1], Yimin Wang[1], Yinan Zhang[1], Xiaofan Jiang[1], Dan Wang[1], Qingtian Ma[1], Huajun Li[3], Caigang Liu[1]*

[1]Cancer Stem Cell and Translation Medicine Lab, Innovative Cancer Drug Research and Development Engineering Center of Liaoning Province, Department of Oncology, Shengjing Hospital of China Medical University, Shenyang, China; [2]Department of Urology Surgery, Shengjing Hospital of China Medical University, Shenyang, China; [3]Clinical Research and Development, Jiangsu Hengrui Pharmaceuticals Co Ltd, Shanghai, China

**Abstract** Recent evidences from clinical trials (NCT04486911) revealed that the combination of pyrotinib, letrozole, and dalpiciclib exerted optimistic therapeutic effect in treating HER2+HR+ breast cancer; however, the underlying molecular mechanism remained elusive. Through the drug sensitivity test, the drug combination efficacy of pyrotinib, tamoxifen, and dalpiciclib to BT474 cells was tested. The underlying molecular mechanisms were investigated using immunofluorescence, Western blot analysis, immunohistochemical staining, and cell cycle analysis. Potential risk factor that may indicate the responsiveness to drug treatment in HER2+/HR+ breast cancer was identified using RNA-sequence and evaluated using immunohistochemical staining and in vivo drug susceptibility test. We found that pyrotinib combined with dalpiciclib exerted better cytotoxic efficacy than pyrotinib combined with tamoxifen in BT474 cells. Degradation of HER2 could enhance ER nuclear transportation, activating ER signaling pathway in BT474 cells, whereas dalpiciclib could partially abrogate this process. This may be the underlying mechanism by which combination of pyrotinib, tamoxifen, and dalpiciclib exerted best cytotoxic effect. Furthermore, CALML5 was revealed to be a risk factor in the treatment of HER2+/HR+ breast cancer and the usage of dalpiciclib might overcome the drug resistance to pyrotinib + tamoxifen due to CALML5 expression. Our study provided evidence that the usage of dalpiciclib in the treatment of HER2+/HR+ breast cancer could partially abrogate the estrogen signaling pathway activation caused by anti-HER2 therapy and revealed that CALML5 could serve as a risk factor in the treatment of HER2+/HR+ breast cancer.

**\*For correspondence:**
angel-s205@163.com

†These authors contributed equally to this work

## Editor's evaluation

This study presents a valuable finding on the combination use of pyrotinib, tamoxifen and dalpiciclib against HER2+/HR+ breast cancer. The evidence supporting the claims of the authors is solid. The work will be of interest to medical biologists or clinicians working on breast cancer.

## Introduction

Human epidermal growth factor receptor 2-positive (HER2+) breast cancer is associated with an increased risk of disease recurrence and death (*Perou et al., 2000*; *Slamon et al., 1987*; *Tzahar et al., 1996*). HER2-overexpressing breast cancers have high heterogeneity, accounting partially for

the co-expression of hormone receptors (HRs) (*Loi et al., 2016*). Previous studies have demonstrated that extensive cross-talk exists between the HER2 signaling pathway and the estrogen receptor (ER) pathway (*Wang et al., 2011*). In addition, exposure to anti-HER2 therapy may reactivate the ER signaling pathway, which could lead to drug resistance (*Brandão et al., 2020*). Generally, however, HER2-positive patients are treated using the same algorithms, both in the early and advanced stages (*Moja et al., 2012*).

Increasing evidence has confirmed that the intrinsic differences between HER2$^+$/HR$^+$ and HER2$^+$/HR$^-$ patients should not be ignored (*Carey et al., 2016*). Clinical outcomes have demonstrated that HER2$^+$/HR$^+$ breast cancer patients have a lower chance of achieving a pathologically complete response than HER2$^+$/HR$^-$ patients, when treated with neoadjuvant chemotherapy plus anti-HER2 therapy (*Cameron et al., 2017*; *Cortazar et al., 2014*). Nevertheless, the addition of concomitant endocrine therapy to anti-HER2 therapy or chemotherapy did not show any advantages in clinical trials, such as the NSABP B-52 and ADAPT HER2$^+$/HR$^+$ studies (*Harbeck et al., 2017*; *Rimawi et al., 2017*). Recently, the synergistic effect of CDK4/6 (cyclin kinase 4/6) inhibitors and anti-HER2 drugs in HER2$^+$ breast cancer has been reported. The combination of anti-HER2 drugs and CDK4/6 inhibitors showed strong synergistic effects and high efficacy in HER2$^+$ breast cancer cells (*Goel et al., 2016*; *Zhang et al., 2019*). Besides, in the recent MUKDEN 01 clinical trial (NCT04486911), the combination use of pyrotinib (anti-HER2 drug), letrozole (endocrine drug), and dalpiciclib (CDK4/6 inhibitor) exerted optimal therapeutic effect in HER2$^+$HR$^+$ breast cancer patients and offered novel chemo-free neoadjuvant therapy for the treatment of HER2$^+$HR$^+$ breast cancer (*Niu et al., 2022*), yet the underlying mechanism warrants further investigation.

Herein, we investigated the underlying molecular mechanism how the combination of pyrotinib, letrzole, and dalpiciclib achieved satisfactory therapeutic effect in MUKDEN 01 trial. We studied the combined effect of pyrotinib (anti-HER2 drug), tamoxifen (endocrine therapy), and dalpiciclib (CDK4/6 inhibitor) on the HER2$^+$/HR$^+$ breast cancer cell line BT474 to simulate the clinical therapy in MUKDEN 01 trial (*Niu et al., 2022*). We found that pyrotinib combined with dalpiciclib exerted better cytotoxic efficacy than pyrotinib combined with tamoxifen. Moreover, the combination use of pyrotinib, tamoxifen, and dalpiciclib displayed the best cytotoxic effect both in vitro and in vivo. In addition, HER2-targeted therapy induced nuclear ER redistribution in HER2$^+$/HR$^+$ cells and the activation of ER signaling pathway, which could be partially abrogated by the addition of dalpiciclib. Furthermore, the expression of CALML5 could be a potential risk factor in the treatment of HER2$^+$HR$^+$ breast cancer and the introduction of dalpiciclib could partially abrogate the drug resistance to pyrotinib + tamoxifen caused by the high expression of CALML5 in HER2$^+$HR$^+$ breast cancer. Our study provided potential molecular mechanisms why the combination of pyrotinib, letrozole, and dalpiciclib could achieve satisfactory clinical response and found CALML5 as a potential risk factor in the treatment of HER2$^+$HR$^+$ breast cancer.

## Results

### Pyrotinib combined with dalpiciclib exerted stronger cytotoxic effect than pyrotinib combined with tamoxifen

To explore the effects of anti-HER2 drugs, tamoxifen, and dalpiciclib in HER2$^+$/HR$^+$ breast cancer, we first evaluated the cytotoxic activities of these three reagents in BT474 breast cancer cells. The results indicated that the IC$_{50}$ doses for pyrotinib, trastuzumab, tamoxifen, and dalpiciclib were 10 nM, 170 µg/ml, 5 µM, and 8 µM, respectively (*Figure 1—figure supplement 1a*). To further investigate whether these drugs could have a synergistic effect on BT474 cells, we assessed the cytotoxic effects of the combinations of pyrotinib and dalpiciclib, pyrotinib and tamoxifen, and tamoxifen and dalpiciclib at different concentrations. We calculated the combination index for each combination using Compusyn software to determine whether the antitumor effects were synergistic (*Chou and Talalay, 1984*). Synergistic effects were observed in the combination group of pyrotinib and dalpiciclib, as well as in the pyrotinib and tamoxifen groups; both with CI values of <1 at several concentrations (*Figure 1a*). However, in the combination group of tamoxifen and dalpiciclib, no synergistic effect was observed.

We also analyzed the effect of the three-drug combination, and it showed a stronger cytotoxic effect on HER2$^+$/HR$^+$ breast cancer compared with the effect of the other two-drug combinations

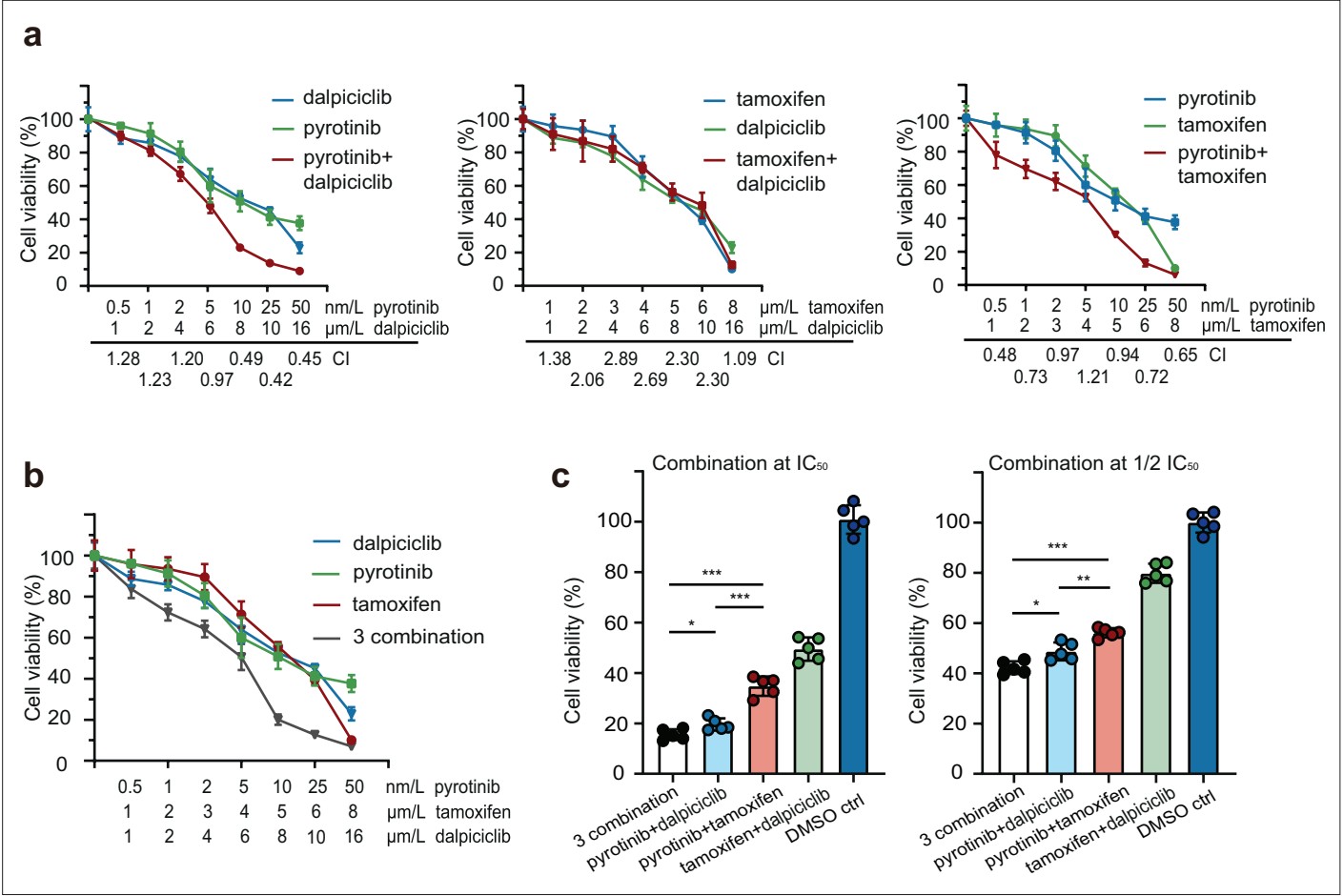

**Figure 1.** Drug sensitivity test of pyrotinib, tamoxifen, dalpiciclib, and their combination on BT474 cells. (**a, b**) Drug sensitivity assay of BT474 cells to single drug and different drug combination. (Data presented as mean ± SDs, all drug sensitivity assay were performed independently in triplicates.) (**c**) Drug sensitivity assay of BT474 cells to different drug combination at $IC_{50}$ concentration and 1/2 $IC_{50}$ concentration. (Data presented as mean ± SDs, *p<0.05, **p<0.01, and ***p<0.001 using Student's *t*-test; all the assays were performed independently in triplicates.) Statistical data is provided in *Figure 1—source data 1*.

The online version of this article includes the following source data and figure supplement(s) for figure 1:

**Source data 1.** Statistical data of *Figure 1*.

**Figure supplement 1.** Colony formation assay of pyrotinib, tamoxifen, dalpiciclib and their combination on BT474 cells.

**Figure supplement 1—source data 1.** Statistical data for *Figure 1—figure supplement 1*.

(*Figure 1b*). As both dalpiciclib and tamoxifen showed synergistic effects in combination with pyrotinib, we sought the combination that exerted better cytotoxic efficacy. Hence, we treated the BT474 cells with different combinations at $IC_{50}$ or half $IC_{50}$ concentrations. The three-drug combination and the combination of pyrotinib and dalpiciclib showed a stronger cell inhibition compared with that exerted by pyrotinib and tamoxifen as well as tamoxifen and dalpiciclib (*Figure 1c*). In the colony formation assay, the three-drug combination group formed the least colonies and the group that was treated by the combination of pyrotinib and dalpiciclib formed the second least colonies. The result of the colony formation assay was consistent to the results of the drug susceptibility test (*Figure 1—figure supplement 1b*).

## Nuclear ER distribution is increased after anti-HER2 therapy and could be partially abrogated by the introduction of dalpiciclib

The results of the drug sensitivity test showed that the combination of pyrotinib and tamoxifen was less effective than the combination of pyrotinib and dalpiciclib on cytotoxic effects. Considering that

the expression of HER2 could affect the distribution of the ER (*Yang et al., 2004*), we performed immunofluorescence staining for ER distribution on the different drug-treated groups to see whether anti-HER2 therapy could degrade HER2 and affect the distribution of ER. We found that pyrotinib induced ER nuclear translocation in BT474 cells, which could be partially abrogated by the addition of dalpiciclib, rather than tamoxifen (*Figure 2a*). Besides trastuzumab, the monoclonal antibody of HER2 could also enhance the nuclear shift of ER and could also be abrogated by the introduction of dalpiciclib (*Figure 2—figure supplement 1c*). Western blot analyses revealed that although the total expression of ER was reduced, the nuclear ER levels increased considerably after the use of pyrotinib (*Figure 2—figure supplement 1a and b*). The use of tamoxifen increased the expression of total ER and nuclear ER (*Figure 2—figure supplement 1a and b*). However, when dalpiciclib was introduced, the increased expression of nuclear ER caused by pyrotinib was partially abrogated (*Figure 2—figure supplement 1b*), and this was consistent with the finding that dalpiciclib could increase the ubiquitination of ER (*Figure 2—figure supplement 1d*).

Based on our in vitro findings, we further explored the ER distribution in clinical samples from the different treatment groups. To this end, we collected the clinical information of HER2+/HR+ patients who received neoadjuvant therapy at the Shengjing Hospital (*Table 1*). We found significant elevations in the nuclear ER expression levels of patients who received chemotherapy(doxetaxel + carboplatin) and anti-HER2 therapy (trastuzumab) compared with the levels in patients who only received chemotherapy (doxetaxel + carboplatin) (*Figure 2b and c*). However, in our clinical trial (NCT04486911, an open-label, multicenter phase II clinical study of pyrotinib maleate combined with CDK4/6 inhibitor and letrozole in neoadjuvant treatment of stage II–III triple-positive breast cancer) (*Niu et al., 2022*), the nuclear ER expression levels of patients did not show significant elevations after the HER2-targeted therapy combined with dalpiciclib (*Figure 2b and c*). These findings verified that the ER receptor may have relocated to the nucleus after anti-HER2 therapy, which could be abrogated with the introduction of dalpiciclib.

## Bioinformatic analyses unravel the synergistic mechanisms underlying the dalpiciclib and pyrotinib in HER2+/HR+ breast cancer

To further explore the mechanisms how dalpiciclib could partially abrogate the activation of ER signaling pathway after pyrotinib treatment, we first analyzed the gene expression profiles of the breast cancer cells treated with pyrotinib via RNA-seq. The signaling pathway enrichment analysis of the differentially expressed genes (DEGs) showed that the majority of the DEGs were significantly enriched in the TNF signaling pathway and cell cycle, while steroid biosynthesis was also strongly active, suggesting that the steroid hormone pathway was activated by pyrotinib (*Figure 3a and b*). Consistent with the findings of signaling pathway enrichment analysis above, Gene Set Enrichment Analysis (GSEA) revealed the following results. The administration of pyrotinib resulted in downregulation of the cell cycle and activation of the hormone pathway. The leading-edge subset of these pathways included the MITOTIC SPINDLE, G2M CHECKPOINT, and ESTROGEN RESPONSE EARLY (*Figure 3c*). These results showed good concordance with our in vitro findings.

We then investigated the alteration of the gene expression profiles between breast cancer cells treated with triple-combined drugs (pyrotinib, tamoxifen, and dalpiciclib) and those treated with the dual-combined drugs (pyrotinib and tamoxifen) via gene enrichment analyses. The results suggested that the addition of dalpiciclib markedly reduced cell cycle progression. This was characterized by the enrichment of the cell cycle and the DNA replication process (*Figure 3e*). The GSEA results further indicated that the progression of the cell cycle was impeded by the enrichment of the gene sets, including MITOTIC SPINDLE and G2M CHECKPOINT (*Figure 3f*).

The activation of the ER pathway might be involved in the effect of pyrotinib on HER2+/HR+ breast cancer cells; therefore, intersection analyses were performed to confirm this. As shown in *Figure 3g*, *CALML5*, *KRT15*, and *KRT19* are the common genes shared between the two sets, the upregulated genes treated with pyrotinib compared to DMSO control group and the genes belonging to the estrogen signaling pathway. Since dalpiciclib is a cell cycle blocker, we also analyzed the common genes involved in the upregulation of the genes and the cell cycle progression after pyrotinib treatment. *CDKN1A* was the only shared gene in these two sets (*Figure 3h*). We then investigated whether any of the abovementioned genes were upregulated with the use of pyrotinib and whether this could be abrogated with the introduction of dalpiciclib, which may serve as a potential risk factor in the

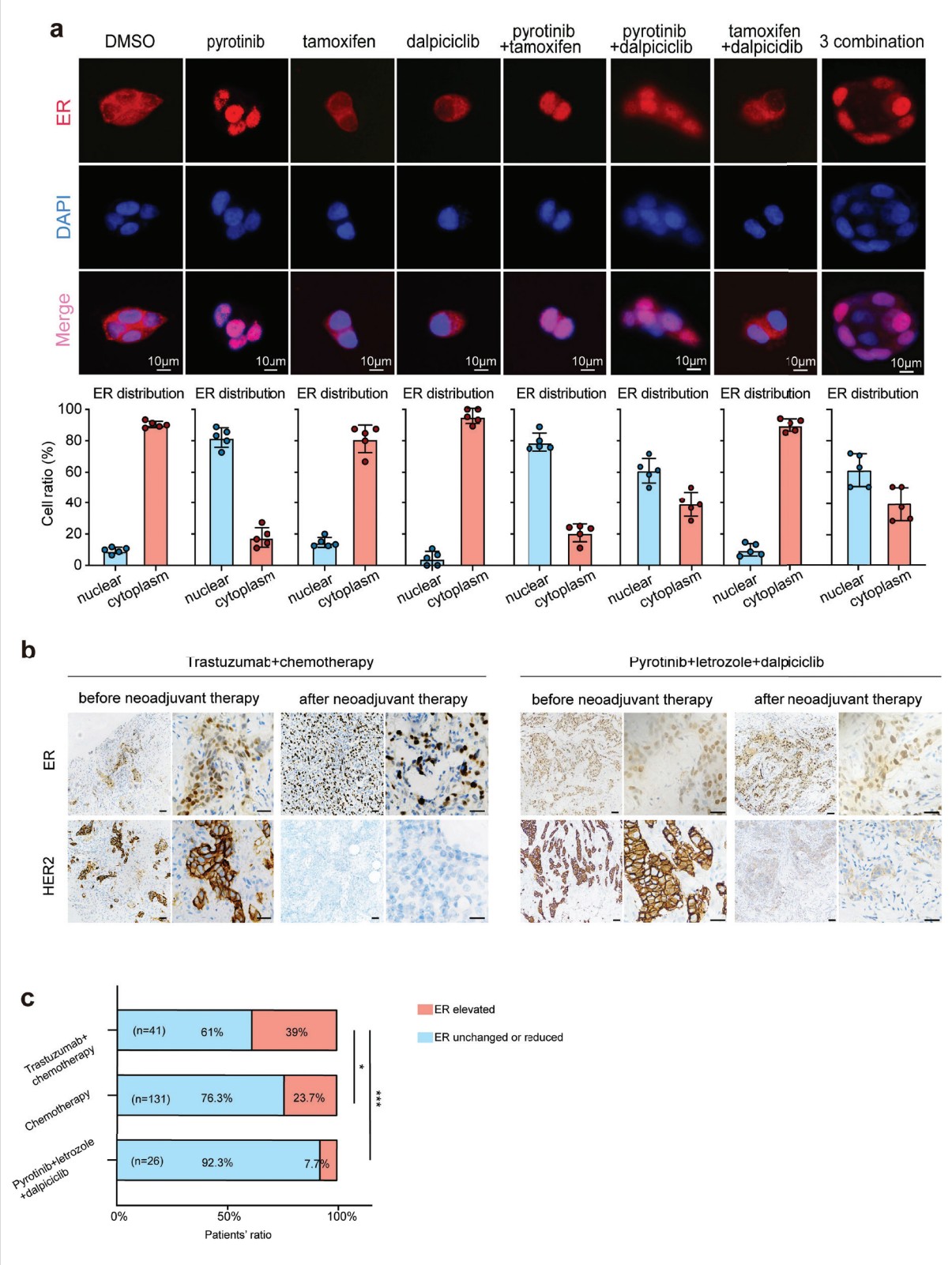

**Figure 2.** Anti-HER2 therapy could lead estrogen receptor (ER) shifting into cell nucleus in HER2[+]/HR[+] breast cancer while CDK4/6 inhibitor could reverse the nuclear translocation of ER. (**a**) Distribution of estrogen receptor in BT474 cell line after different drug (pyrotinib, tamoxifen, and dalpiciclib) treatment. (The distribution ratio of ER was calculated manually by randomly chosen five views in ×400 magnification. All the assays were performed independently in triplicates.) (**b**) Representative views of ER and HER2 expression in patients before and after anti-HER2 (trastuzumab) + chemotherapy

*Figure 2 continued on next page*

*Figure 2 continued*

(docetaxel + carboplatin) and representative views of ER and HER2 expression in patients before and after pyrotinib + letrozole + dalpiciclib treatment. (**c**) Ratio of patients with elevated ER expression and patients with unchanged or reduced ER expression in different kinds of neoadjuvant therapy groups. (***p<0.001 using chi-square test.) Statistical data is provided in *Figure 2—source data 1*.

The online version of this article includes the following source data and figure supplement(s) for figure 2:

**Source data 1.** Statistical data of *Figure 2*.

**Figure supplement 1.** Dalpiciclib partially abrogates ER nuclear transportation induced by anti-HER2 therapy.

**Figure supplement 1—source data 1.** Original gels for *Figure 2—figure supplement 1a, b, and d*.

**Figure supplement 1—source data 2.** Statistical data for *Figure 2—figure supplement 1*.

treatment of HER2+HR+ breast cancer. The results showed that only one factor, *CALML5*, was the common gene (*Figure 3i*).

## CALML5 is a potential risk factor in the treatment of HER2+HR+ breast cancer

Western blot analyses and bioinformatic analyses were conducted to verify the changes in the signaling pathways. The Western blot analyses showed that despite the introduction of tamoxifen partially inhibited the HER2 downstream pathway (AKT-mTOR signaling pathway), it did not significantly affect the phosphorylation of Rb (*Figure 4a*). In contrast, the combination of pyrotinib and

**Table 1.** Demographic information of HER2+/HR+ breast cancer patients who received neoadjuvant therapy.

| Variables | Chemotherapy | Chemotherapy + trastuzumab | Pyrotinib + dalpiciclib + letrozole | p-value |
|---|---|---|---|---|
| No. of patients | 131 | 41 | 26 | |
| *Age (years)* | | | | ns |
| ≤50 | 82 (62.60) | 25 (61.00) | 16 (61.53) | |
| >50 | 49 (37.40) | 16 (39.00) | 10 (38.47) | |
| *T stage* | | | | ns |
| 1 | 15 (11.45) | 5 (12.20) | 2 (7.70) | |
| 2 | 90 (68.70) | 32 (78.04) | 21 (80.76) | |
| 3 | 26 (19.85) | 4 (9.76) | 3 (11.54) | |
| *ER status* | | | | ns |
| ≤30% | 31 (23.66) | 8 (19.51) | 2 (7.6) | |
| >30% | 100 (76.34) | 33 (80.49) | 24 (92.4) | |
| *PR status* | | | | ns |
| ≤30% | 80 (61.07) | 15 (36.59) | 13 (50) | |
| >30% | 51 (38.93) | 26 (63.41) | 13 (50) | |
| *HER2 status* | | | | ns |
| (++) | 78 (59.54) | 12 (29.27) | 10 (38.5) | |
| (+++) | 53 (40.46) | 29 (70.73) | 16 (61.5) | |
| *Ki67 index* | | | | ns |
| <20% | 51 (38.93) | 16 (39.00) | 8 (30.8) | |
| >20% | 80 (61.07) | 25 (61.00) | 18 (69.2) | |

ns, nonsignificant; PR, partial response; ER, estrogen receptor.

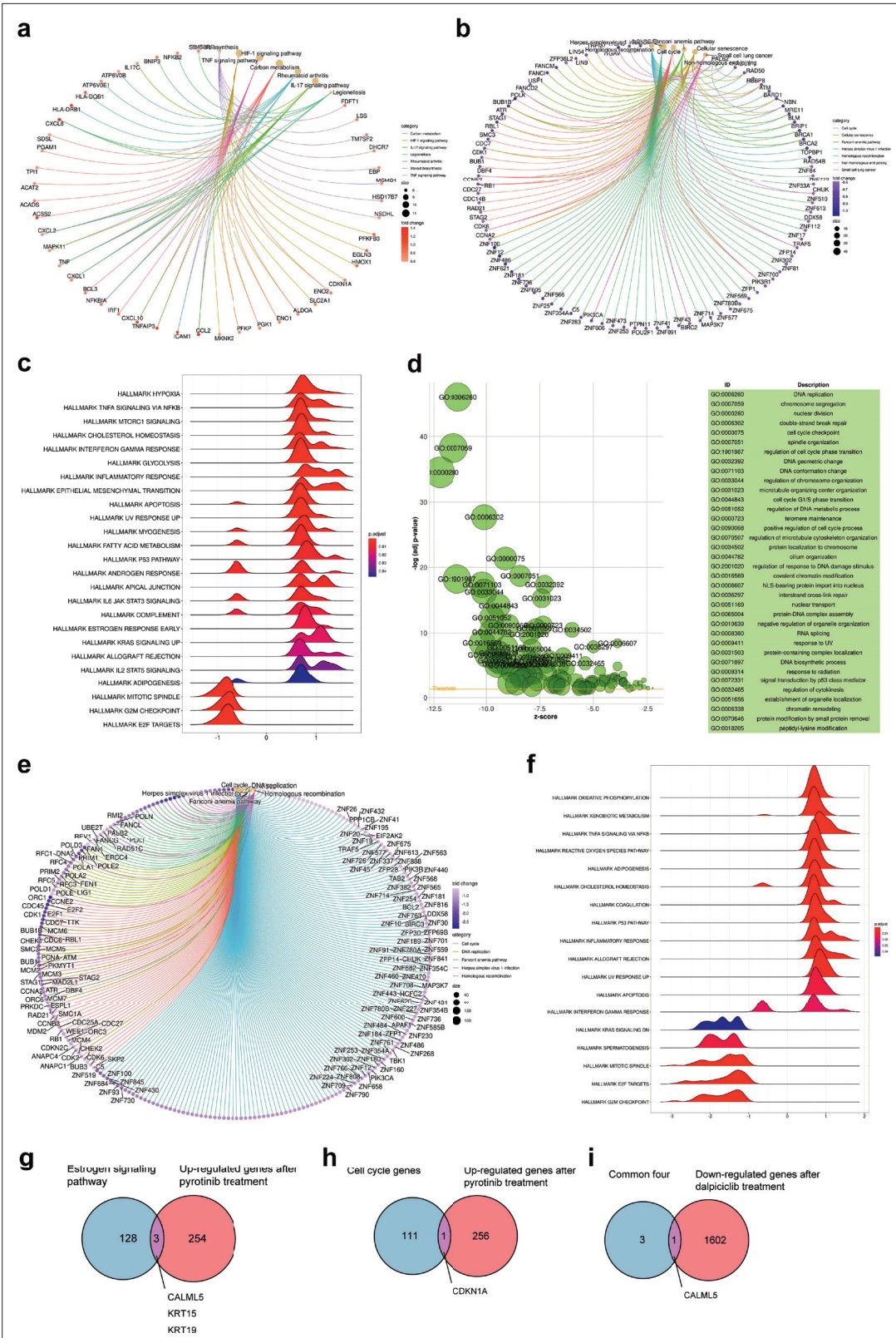

**Figure 3.** Bioinformatic analysis revealed dalpiciclib and pyrotinib blocking HER2 pathway and cell cycle in BT474 cells synergistically. (**a, b**) Signaling pathway enrichment analysis of mRNA changes of BT474 cells treated with pyrotinib compared to BT474 cells treated with 0.1% DMSO. (**c**) Gene Set Enrichment Analysis (GSEA) of mRNA changes of BT474 cells treated with pyrotinib compared to BT474 cells treated with 0.1% DMSO. (**d, e**) Signaling pathway enrichment analysis of mRNA changes of BT474 cells treated with pyrotinib + tamoxifen + dalpiciclib compared to BT474 cells treated with

*Figure 3 continued on next page*

*Figure 3 continued*

pyrotinib + tamoxifen. (**f**) GSEA of mRNA changes of BT474 cells treated with pyrotinib + tamoxifen + dalpiciclib compared to BT474 cells treated with pyrotinib + tamoxifen. (**g**) Intersection of genes that was upregulated after pyrotinib treatment and belonged to estrogen receptor signaling pathway (genes belonging to estrogen receptor signaling pathway are provided in *Figure 3—source data 1*). (**h**) Intersection of genes that were upregulated after pyrotinib treatment and belonged to cell cycle genes (genes belonged to cell cycle gens are provided in *Figure 3—source data 2*). (**i**) Intersection of the four genes that were upregulated after pyrotinib treatment and were downregulated after the introduction of dalpiciclib (genes that were upregulated after pyrotinib treatment and were downregulated after the introduction of dalpiciclib are provided in *Figure 3—source data 3* and *Figure 3—source data 4*).

The online version of this article includes the following source data for figure 3:

**Source data 1.** Gene list in estrogen receptor (ER) signaling pathway summarized by KEGG database for *Figure 3g*.

**Source data 2.** Gene list in cell cycle genes summarized by KEGG database for *Figure 3h*.

**Source data 3.** Upregulated genes after pyrotinib treatment compared to DMSO treatment for *Figure 3g and i*.

**Source data 4.** Downregulated genes after dalpiciclib treatment compared to DMSO treatment for *Figure 3h and i*.

dalpiciclib showed similar inhibition of HER2 downstream pmTOR as the combination of pyrotinib and tamoxifen (*Figure 4a*). However, the combination of pyrotinib and dalpiciclib significantly reduced pRb expression and pCDK4(Thr172) expression (*Figure 4a*). In addition, cell arrest analyses of the different drug combinations were performed. As shown in *Figure 4b*, compared with the cells treated with pyrotinib or tamoxifen, the introduction of dalpiciclib significantly increased the number of cells arrested in the G1/S phase. This confirmed the synergistic inhibition of cell proliferation by dalpiciclib and pyrotinib.

To verify whether CALML5 could be a potential risk factor of treatment responsiveness in clinical practice, clinical samples were collected from HER2⁺/HR⁺ patients before and after neoadjuvant therapy (anti-HER2 therapy (trastuzumab) + chemotherapy (docetaxel + carboplatin) or anti-HER2 therapy (pyrotinib) + CDK4/6 inhibitor (dalpiciclib) + endocrine therapy (letrozole)) (*Table 2*). Immunohistochemical staining of CALML5 showed that the CALML5-positive cells indicated worse drug sensitivities and lower probabilities of achieving pathological complete response (pCR) and partial response (PR) in patients receiving neoadjuvant therapy (*Figure 4c*). However, pyrotinib + letrozole + dalpiciclib displayed better pCR and PR rates than trastuzumab + chemotherapy (docetaxel + carboplatin) in patients with CALML5-positive cells (*Figure 4c*). Moreover, the positive rate of CALML5 decreased after pyrotinib + letrozole + dalpiciclib treatment (*Figure 4d*), consistent with the results of the bioinformatic analyses. Furthermore, xenografts models derived from BT474 cells were also used to test the fuction of CALML5 in models using pyrotinib + tamoxifen or pyrotinib + tamoxifen + dalpiciclib. After knock down *CALML5* (*Figure 4—figure supplement 1a*), the tumor seemed to be more sensitive to the treatment of pyrotinib + tamoxifen (*Figure 4e and f*), and it showed similar response compared to the group treated with three-drug combination (*Figure 4e and f*). Hence, using clinical specimens as well as in vivo models, we found that the expression of CALML5 might be the potential risk factor in the treatment of HER2⁺HR⁺ breast cancer and the introduction of dalpiciclib could abrogate the drug resistance to pyrotinib + tamoxifen in HER2⁺HR⁺ breast cancer due to the expression of CALML5.

## Discussion

Until now, the combination of anti-HER2 therapy and chemotherapy has been the major treatment strategies for treatment of HER2⁺/HR⁺ breast cancer (*Gianni et al., 2012*; *Schneeweiss et al., 2013*). Although pCR and DFS improve with the use of the combination of anti-HER2 therapy and chemotherapy, the strong adverse effects of chemotherapy cannot be ignored (*Maguire et al., 2021*). Moreover, clinical data showed that the addition of anti-estrogen receptor drugs in the treatment regimen of HER2⁺/HR⁺ breast cancer did not provide additional advantages in the pCR rates and DFS (*Harbeck et al., 2017*; *Rimawi et al., 2017*). Hence, with the rapid development of small-molecule drugs such as tyrosine kinase inhibitors (TKIs) and CDK4/6 inhibitors, additional chemo-free strategies are being developed for the treatment of HER2⁺/HR⁺ breast cancer (*Gianni et al., 2018*; *Pascual et al., 2021*; *Saura et al., 2014*). In the recent MUKDEN 01 clinical trial (NCT04486911), the combination of pyrotinib, letrozole, and dalpiciclib achieved satisfactory clinical response in HER2⁺HR⁺ patients with minimal adverse effects and offered novel chemo-free neoadjuvant therapy for HER2⁺HR⁺ patients

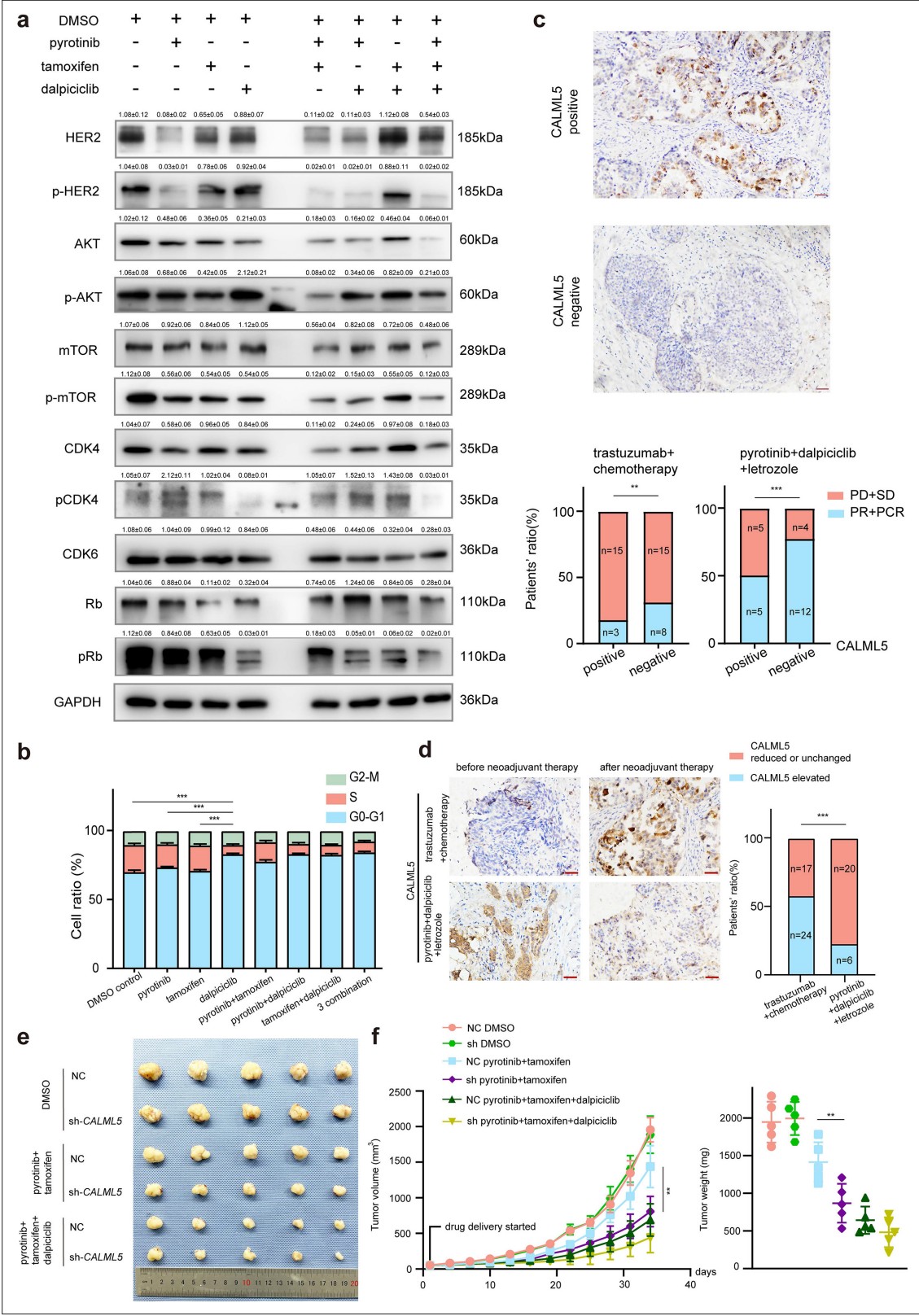

**Figure 4.** CALML5 could serve as a potential risk factor in the treatment of HER2[+]HR[+] breast cancer. (**a**) Western blot analysis of HER2 signaling pathway and cell cycle pathway in BT474 cells treated with different drugs or their combination. (This assay was performed in triplicates independently.) (**b**) Cell cycle analysis in BT474 cells treated with different drugs or their combination. (Data presented as mean ± SDs, ***p<0.001 using Student's *t*-test; all the assays were performed independently in triplicates.) (**c**) Representative views of CALML5 positive/negative tissue. The difference of PR + PCR ratio

*Figure 4 continued on next page*

*Figure 4 continued*

and PD + SD ratio in patients who received anti-HER2 therapy (trastuzumab) + chemotherapy (docetaxel + carboplatin) or pyrotinib + dalpiciclib + letrozole regarding on their expression of CALML5. (***p<0.001 using chi-square test.) (**d**) Representative views of CALML5 positive/negative tissue. Ratio of patients with elevated or decreased CALML5 after receiving anti-HER2 therapy (trastuzumab) + chemotherapy (docetaxel + carboplatin) or pyrotinib + dalpiciclib + letrozole. (***p<0.001 using chi-square test.) (**e**) Representative views of xenograft tumors derived from BT474 NC (NC stands for negative control) or BT474 sh cell lines treated with different drug combination. (***p<0.001 using Student's *t*-test.) (**f**) Growth curves and tumor weight of xenograft tumors derived from BT474 NC or BT474 sh cell lines treated with different drug combination. (n = 5 in each group, ***p<0.001 using Student's *t*-test.) Raw gels are provided in *Figure 4—source data 1*, statistical data is provided in *Figure 4—source data 2*, and original files of cell cycle analysis are provided in *Figure 4—source data 3*.

The online version of this article includes the following source data and figure supplement(s) for figure 4:

**Source data 1.** Original files for the gels in *Figure 4a*.

**Source data 2.** Histograms of the cell cycle analysis in *Figure 4b*.

**Source data 3.** Statistical data for *Figure 4*.

**Figure supplement 1.** The expression of CALML5 was elevated in HER2+HR+ breast cancer due to pyrotinib treatment and theelevation of CALML5 could be abrogated by dalpiciclib.

**Figure supplement 1—source data 1.** Statistical data for *Figure 4—figure supplement 1*.

**Table 2.** Demographic information of HER2$^+$/HR$^+$ breast cancer patients who were evaluated for CALML5 before receiving neoadjuvant therapy.

| Variables | Chemotherapy + trastuzumab | Pyrotinib + dalpiciclib + letrozole | p-value |
|---|---|---|---|
| No. of patients | 41 | 26 | |
| *Age (years)* | | | |
| ≤50 | 25 (61.00) | 16 (61.53) | ns |
| >50 | 16 (39.00) | 10 (38.47) | |
| *T stage* | | | |
| 1 | 5 (12.20) | 2 (7.70) | ns |
| 2 | 32 (78.04) | 21 (80.76) | |
| 3 | 4 (9.76) | 3 (11.54) | |
| *ER status* | | | |
| ≤30% | 8 (19.51) | 2 (7.6) | 0.0145 |
| >30% | 33 (80.49) | 24 (92.4) | |
| *PR status* | | | |
| ≤30% | 15 (36.59) | 13(50) | ns |
| >30% | 26 (63.41) | 13(50) | |
| *HER2 status* | | | |
| (++) | 12 (29.27) | 10 (38.5) | ns |
| (+++) | 29 (70.73) | 16 (61.5) | |
| *Ki67 index* | | | |
| <20% | 16 (39.00) | 8 (30.8) | ns |
| >20% | 25 (61.00) | 18 (69.2) | |
| CALML5 | | | |
| positive | 18 (43.90) | 10 (38.46) | ns |
| negative | 23 (56.10) | 16 (43.9) | |

ns, nonsignificant; PR, partial response; ER, estrogen receptor.

(*Niu et al., 2022*). The molecular mechanism of how the combination of pyrotinib, letrozole and dalpiciclib achieved optimal therapeutic effect remained elusive.

In our study, we found that the combination of pyrotinib and dalpiciclib exerted stronger cytotoxic effects on BT474 cells than the combination of pyrotinib and tamoxifen. This was anomalous since the two blocking agents of HER2 and ER were expected to inhibit their crosstalk and achieve better responses. To explore the potential mechanisms, we investigated the crosstalk between HER2 and the ER. After degrading HER2 with pyrotinib, ER was found to relocate to the cell nucleus, enhancing the function of ER, which was consistent with the findings of *Kumar et al., 2002* and *Yang et al., 2004*. We believe that the anti-HER2-mediated ER redistribution caused the enhanced ER function, leading to the relatively low cytotoxic efficacy of the combination of pyrotinib and tamoxifen in the treatment of HER2+/HR+ cells. Moreover, we found that the introduction of dalpiciclib to pyrotinib significantly decreased the total and nuclear expression of ER, and partially abrogated the ER activation caused by pyrotinib. This may be the underlying mechanism by which the addition of dalpiciclib could achieve better response in the in vitro and in vivo studies.

Furthermore, using mRNA-seq and bioinformatics analyses, CALML5 was identified as a potential risk factor in the treatment of HER2+HR+ breast cancer. CALML5, known as calmodulin-like 5, is a skin-specific calcium-binding protein that is closely related to keratinocyte differentiation (*Méhul et al., 2001*). A previous study showed that the high expression of CALML5 was strongly associated with better survival in patients with head and neck squamous cell carcinomas (*Wirsing et al., 2021*). *Misawa et al., 2020* reported that the methylation of CALML5 led to its downregulation, and this showed a correlation with HPV-associated oropharyngeal cancer. Moreover, the ubiquitination of CALML5 in the nucleus was found to play a role in the carcinogenesis of breast cancer in premenopausal women (*Debald et al., 2013*). Our results suggested that the high expression of CALML5 in HER2+HR+ breast cancer patients may lead to the resistant of pyrotinib combined with tamoxifen and the introduction of dalpiciclib might overcome this drug resistance and offer better therapeutic effects. However, the underlying mechanism of CALML5 in breast cancer warrants further investigation.

In conclusion, our study investigated the underlying synergistic mechanism for the combination of pyrotinib, letrozole, and dalpiciclib in the MUKDEN 01 clinical trial (NCT04486911). We identified the novel role of the dalpiciclib in HER2+/HR+ breast cancer, provided evidence that CALML5 may serve as a potential risk factor in the treatment of HER2+HR+ breast cancer and the introduction of dalpiciclib might overcome the drug resistance to pyrotinib + tamoxifen due to the expression of CALML5 in HER2+HR+ breast cancer.

# Materials and methods

**Key resources table**

| Reagent type (species) or resource | Designation | Source or reference | Identifiers | Additional information |
|---|---|---|---|---|
| Cell line (BT474) | HER2+/HR+ breast cancer cell line | ATCC | | Cell line cultured in RMPI 1640 Culture medium supplemented with 10% FBS |
| Transfected construct (human) | CALML5 shRNA #1,2,3 | Genechem Technologies | Cat# GIEL0313139 | Lentiviral construct to transfect and express the shRNA |
| Antibody | Anti-ER (rabbit polyclonal) | CST | Cat #13258 | IF (1:400), WB (1:1000) |
| Antibody | Anti-pHER2(Tyr 1221/1222, rabbit polyclonal) | CST | Cat #2243 | WB (1:1000) |
| Antibody | Anti-HER2 (rabbit polyclonal) | CST | Cat #4290 | WB (1:1000) |
| Antibody | Anti-pAKT (Ser473, rabbit polyclonal) | CST | Cat #4060 | WB (1:2000) |
| Antibody | Anti-AKT (rabbit polyclonal) | CST | Cat #4685 | WB (1:1000) |
| Antibody | Anti-pmTOR (Ser2448, rabbit polyclonal) | CST | Cat #5536 | WB (1:1000) |

*Continued on next page*

*Continued*

| Reagent type (species) or resource | Designation | Source or reference | Identifiers | Additional information |
|---|---|---|---|---|
| Antibody | Anti-mTOR (rabbit polyclonal) | CST | Cat #2983 | WB (1:1000) |
| Antibody | Anti-pRb (Ser780, rabbit polyclonal) | CST | Cat #8180 | WB (1:1000) |
| Antibody | Anti-Rb (rabbit polyclonal) | CST | Cat #9309 | WB (1:2000) |
| Antibody | Anti-CDK4 (rabbit polyclonal) | CST | Cat #12790 | WB (1:1000) |
| Antibody | Anti-CDK6 (rabbit polyclonal) | CST | Cat #13331 | WB (1:1000) |
| Antibody | Anti-Ubi (mouse monoclonal) | CST | Cat #3936 | WB (1:1000) |
| Antibody | Anti-Lamin A (mouse monoclonal) | CST | Cat #4777 | WB (1:2000) |
| Antibody | Anti-HSP90 (mouse monoclonal) | CST | Cat #4877 | WB (1:1000) |
| Antibody | Anti-GAPDH (rabbit monoclonal) | CST | Cat #5174 | WB (1:1000) |
| Antibody | Anti-pCDK4 (Thr172, rabbit polyclonal) | absin | Cat abs139836 | WB (1:1000) |
| Antibody | Anti-ER (rabbit monoclonal) | Abcam | Cat ab32063 | IHC (1:400) |
| Antibody | Anti-HER2 (rabbit monoclonal) | Abcam | Cat ab134182 | IHC (1:400) |
| Antibody | Anti-CALML5 (rabbit polyclonal) | Proteintech | Cat 13059-1-AP | IHC (1:400) |
| Sequence-based reagent | CALML5_F | This paper | PCR primers | CACCATCAATGCCCAGGAGCTG |
| Sequence-based reagent | CALML5_R | This paper | PCR primers | GTCGCTGTCAACCTCGGAGATG |
| Chemical compound, drug | Tamoxifen | MCE | Cat HY-13757A | |
| Software, algorithm | SPSS | SPSS | SPSS, version 22 Armonk, NY, USA | |

## Clinical specimens

A total of 198 HR$^+$/HER2$^+$ patients who received neoadjuvant therapy were enrolled in this study to evaluate the status of ER and CALML5, of which 26 patients were from the clinical trial (NCT04486911, an open-label, multicenter phase II clinical study of pyrotinib maleate combined with CDK4/6 inhibitor and letrozole in neoadjuvant treatment of stage II–III triple-positive breast cancer), 41 patients received anti-HER2 therapy (trastuzumab) + chemotherapy(docetaxel + carboplatin) and 131 patients only received chemotherapy (docetaxel + carboplatin). The demographic data of these clinical patients is displayed in *Table 1* and *Table 2*. The latter two columns of patients in *Table 1* were identical to the patients displayed in *Table 2*. To make the origins of our clinical specimens clearer, we additionally created *Table 2* to explain where the specimens that evaluated CALML5 were from.

The study was approved by the Institutional Ethics Committee and complied with the principles of the Declaration of Helsinki and Good Clinical Practice guidelines of the National Medical Products Administration of China. Informed consent was obtained from all the participants.

## Cell lines and cell cultures

BT474 were purchased from the American Type Culture Collection (ATCC, Manassas, VA), and its identity had been authenticated using STR profiling. The human HER2$^+$/HR$^+$ breast cancer cell line BT474 was cultured in RPMI1640 culture medium supplemented with 10% fetal bovine serum (FBS) and was not contaminated by mycoplasma or other microbiomes.

## Chemicals and antibodies

Pyrotinib (SHR1258) and dalpiciclib (SHR6390) were kindly provided by Hengrui Medicine Co., Ltd. Tamoxifen (HY-13757A) and trastuzumab were purchased from MCE company. Compounds were dissolved in dimethylsulfoxide (DMSO) at a concentration of 10 mM and stored at –20°C for further use. Trastuzumab were dissolved and used according to the manufacturer's instructions. The following antibodies were purchased from Cell Signaling Technology (Beverly, MA): ER (#13258), p-HER2 (Tyr 1221/1222, #2243), HER2 (#4290), p-Akt (Ser473, #4060), AKT (#4685), p-mTOR (Ser2448, #5536), mTOR (#2983), pRb (Ser 780, #8180), Rb (#9309), CDK4 (#12790), CDK6 (#13331), Ubi (#3936), Lamin A (#4777), HSP90 (#4877), and GAPDH (#5174). The pCDK4 (Thr172, abs139836) antibody was purchased from Absin Technologies (Shanghai, China).

## Cell viability assays and drug combination studies

CCK cell viability assays were (Cofitt Life Science) used to quantify the inhibitory effect of the different treatments. Cells were seeded in 96-well plates at a density of 5000 cells/well and treated the next day with DMSO, pyrotinib, trastuzumab, tamoxifen, dalpiciclib, or both drugs in combination for 48 hr. The combination index (CI) values of different drugs were calculated using CompuSyn (ComboSyn Inc). The CI values demonstrated synergistic (<1), additive (1–1.2), or antagonistic (>1.2) effects of the two-drug combinations. The drug sensitivity experiments were performed three times independently.

## Cell cycle analyses

The cells were starved in culture medium supplemented with 2% serum for 24 hr before treatment. Treatments included DMSO (0.1%), pyrotinib (10 nM), dalpiciclib (8 μM), tamoxifen (5 μM), or different combinations of drugs. After treatment for 24 hr, cells in different treating groups were trypsinized, washed with PBS, fixed in 70% ethanol, and incubated overnight at 4°C. Next day, cells were collected, washed, and resuspended in PBS at a concentration of $5 \times 10^5$ cells/mL. The cell solutions were then incubated with a RNase and propidium iodide (PI) solution for 30 min at room temperature without exposure to light and analyzed using a flow cytometer (BD FACS Calibur) according to the manufacturer's instructions. This assay was performed in triplicates.

## Colony formation assays

Cells were seeded in 6-well plates at a density of 1000 cells/well. The cells were treated with DMSO (0.1%), pyrotinib (10 nM), tamoxifen (5 μM), dalpiciclib (8 μM), or a combination of the two or three agents. During the process, the culture medium was renewed every 3 days. After 14 days, the colonies were fixed and stained with crystal violet. Clusters of more than eight cells were counted as colonies. This assay was performed in triplicates independently.

## Western blot analysis

Cells were lysed using a cell lysis buffer (Beyotime, Shanghai, China). The total proteins were extracted in a lysis buffer (Beyotime), and the nuclear proteins were extracted using a nuclear protein extraction kit (Beyotime), in which protease inhibitor (HY-K0010; MCE) and phosphatase inhibitor (HY-K0021; MCE) were added. Protein concentrations were determined using a Pierce BCA Protein Assay Kit (Thermo Fisher Scientific, Waltham, MA) according to the manufacturer's instructions. The proteins from the cells and tissue lysates were separated using 10% SDS-PAGE and 6% SDS-PAGE, respectively, and then transferred to polyvinylidene fluoride (PVDF) membranes. The immunoreactive bands were detected using enhanced chemiluminescence (ECL). The Western blot analysis was performed in triplicates independently.

## Co-immunoprecipitation assay

BT474 cells treated with different drugs were lysed using a cell lysis buffer (Beyotime). in which protease inhibitor (HY-K0010; MCE) and phosphatase inhibitor (HY-K0021; MCE) were added. Protein

concentrations were determined using a Pierce BCA Protein Assay Kit (Thermo Fisher Scientific) according to the manufacturer's instructions. Lysates were clarified by centrifugation, incubated with primary ER antibodies (#8644; Cell Signaling Technologies) overnight at 4°C, and incubated with protein A/G coupled sepharose beads (L1721; Santa Cruz Biotechnology) for 2 hr at 4°C. Bound complexes were washed three times with cell lysis buffer and eluted by boiling in SDS loading buffer. Bound proteins were detected on 6% SDS-PAGE followed by immunoblotting. The immunoreactive bands were detected using enhanced chemiluminescence (ECL).

## Immunofluorescence assays
The cellular localization of different proteins was detected using immunofluorescence. Briefly, the cells grown on glass coverslips were fixed in 4% paraformaldehyde at room temperature for 30 min. Cells were incubated with the respective primary antibodies for 1 hr at room temperature, washed in PBS, and then incubated with 590-Alexa-(red) secondary antibodies (Molecular Probes, Eugene, OR). We used 590-Alexa-phalloidin to localize the ER. The nuclei of the cells were stained with DAPI and color-coded in blue. The images were captured using an immunofluorescence microscope (Nikon Oplenic Lumicite 9000). The distribution ratio of ER was calculated manually by randomly chosen five views in ×400 magnification. The immunofluorescence assay was performed in triplicates independently.

## Immunohistochemical staining
The clinical samples were fixed in 4% formaldehyde, embedded in paraffin, and sectioned continuously at a thickness of 3 µm. The paraffin sections were deparaffinized with xylene and rehydrated using a graded ethanol series. They were then washed with tris-buffered saline (TBS). After these preparation procedures, the sections of each sample were incubated with the primary anti-ER antibody (Abcam Company, ab32063), anti-HER2 antibody (Abcam Company, ab134182), and anti-CALML5 antibody (Proteintech, 13059-1-AP) at 4°C overnight. The next day, they were washed three times with TBS and incubated with a horseradish peroxidase (HRP)-conjugated secondary antibody (Gene Tech Co. Ltd., Shanghai, China) at 37°C for 45 min, followed by immunohistochemical staining using a DAB kit (Gene Tech Co. Ltd.) for 5–10 min.

## Evaluation of the ER and HER2 status
The ER and HER2 statuses of patients who received neoadjuvant therapy were evaluated by a pathologist from a Shenjing-affiliated hospital. The clinical specimens before and after the neoadjuvant therapy were evaluated. The analyses of the elevation or decline in ER statuses were based on these pathological reports. The 2+ of HER2 was detected by immunohistochemistry as well as a FISH test-positive report.

## mRNA-seq and differential gene expression analysis
BT474 cells were treated with 1% DMSO, pyrotinib (10 nM), tamoxifen (5 µM), dalpiciclib (8 µM), pyrotinib + tamoxifen, pyrotinib + dalpiciclib, tamoxifen + dalpiciclib, and combination of three drugs, respectively. Each group was performed in triplicate an treated with drugs for 48 hr. After the treatment, the mRNAs in these cells were extracted using RNAiso Plus (Takara, Cat# 9109) and then sequenced by Biomarker Techonologies using Illumina sequencing technology. The differential gene expression analysis was performed using online tools (http://www.biomarker.com.cn/biocloud), and differentially expressed genes were defined as Log2 Foldchange > 0.5, p-value <0.05. As for the gene set of estrogen signaling pathway and cell cycle genes, genes sets were downloaded from KEGG database.

## Gene enrichment analysis
Gene annotation data in the GO and KEGG databases and R language were used for the enrichment analysis. Only enrichment with q-values < 0.05 were considered significant.

## GSEA
The hallmark gene sets in the Molecular Signatures Database were used for performing the GSEA; only gene sets with q-values < 0.05 were considered significantly enriched.

## Stably knock down of CALML5 in BT474 cell line

The sh-CALML5 lentivirus was synthesized by Genechem Technologies. BT474 cells were cultured in a 6-well plate and transduced with shRNAs targeting human CALML5 or NC (negative control). The sequences for sh-*CALML5* were 5'-ACGAGGAGTTCGCGAGGAT-3' (sequence 1), 5'- AAATCAGC TTCCAGGAGTT-3' (sequence 2), and 5'-GAAACTCATCTCCGAGGTT-3' (sequence 3). The sequence for sh-NC was 5'-GCAGTGAAAGATGTAGCCAAA-3'. The primer sequence for CALML5 was CACC ATCAATGCCCAGGAGCTG (forward) and GTCGCTGTCAACCTCGGAGATG (reverse).

## Animal studies

Four- to five-week-old female NOD scid mice were maintained in the animal husbandry facility of a specific pathogen-free (SPF) laboratory. All experiments were performed in accordance with the Regulations for the Administration of Affairs Concerning Experimental Animals and were approved by the Experimental Animal Ethics Committee of the China Medical University.

Subcutaneous injections of $1 \times 10^7$ BT474 NC cells or BT474 sh-*CALML5* cells were performed to induce tumors. Two weeks after tumor cell inoculation, tumor volume was measured every 3 days and calculated as V = 1/2 (width$^2$ ×length).

As for drug sensitivity test, pyrotinib, tamoxifen, and dalpiciclib were administrated when after 2 weeks of tumor inoculation. Mice inoculated with BT474 NC or BT474 sh-*CALML5* cells were randomly assigned to one of three groups (n = 5 each, total number = 30). Mice carried xenograft tumors were treated by intraperitoneal injection for 28 days with vehicle (1% DMSO dissolved in normal saline/2 days), pyrotinib (20 mg/kg every 3 days), tamoxifen (25 mg/kg every 3 days), and dalpiciclib (75 mg/kg every half a week). When the drug was continuously delivered for 32 days, mice were humanely euthanized and tumors were dissected and analyzed.

## Statistical analysis

All the descriptive statistics were presented as the means ± standard deviations (SDs). The differences between two groups were analyzed by Student's *t*-tests. The differences between percentage data were analyzed using chi-square test. The statistical analyses were performed using IBM SPSS version 22 (SPSS, Armonk, NY) and GraphPad Prism version 7. The statistical significance of the differences between the test and control samples was assessed at significance thresholds of *$p<0.05$, **$p<0.01$, and ***$p<0.001$.

## Acknowledgements

This study was supported by the National Natural Science Foundation of China (#U20A20381, #81872159).

## Additional information

### Competing interests

Huajun Li: is affiliated with Jiangsu Hengrui Pharmaceuticals Co. Ltd and the author has no other competing interests to declare. Caigang Liu: Reviewing editor, eLife. The other authors declare that no competing interests exist.

### Funding

| Funder | Grant reference number | Author |
| --- | --- | --- |
| National Natural Science Foundation of China | U20A20381 | Caigang Liu |
| National Natural Science Foundation of China | 81872159 | Caigang Liu |

The funders had no role in study design, data collection and interpretation, or the decision to submit the work for publication.

## Author contributions
Jiawen Bu, Conceptualization, Data curation, Methodology, Writing - original draft; Yixiao Zhang, Conceptualization, Methodology, Writing - review and editing; Nan Niu, Yinan Zhang, Resources, Validation; Kewei Bi, Data curation, Software; Lisha Sun, Formal analysis, Investigation; Xinbo Qiao, Qingtian Ma, Data curation, Investigation; Yimin Wang, Investigation, Methodology; Xiaofan Jiang, Formal analysis, Methodology; Dan Wang, Conceptualization, Data curation; Huajun Li, Resources; Caigang Liu, Conceptualization, Funding acquisition

## Author ORCIDs
Jiawen Bu ![ORCID] http://orcid.org/0000-0002-7168-3721
Lisha Sun ![ORCID] http://orcid.org/0000-0002-4095-5026
Xinbo Qiao ![ORCID] http://orcid.org/0000-0002-6759-921X
Caigang Liu ![ORCID] http://orcid.org/0000-0003-2083-235X

## Ethics
The animal study was approved by the Ethics Committee of Shengjing Hospital of China Medical University (Permit Number: 2020PS318K). The pdf permission document have been uploaded as a Supporting Zip Document.

## Decision letter and Author response
Decision letter https://doi.org/10.7554/eLife.85246.sa1
Author response https://doi.org/10.7554/eLife.85246.sa2

---

# Additional files

## Supplementary files
• MDAR checklist

## Data availability
Sequencing data have been deposited in GSA database (https://ngdc.cncb.ac.cn/) under accession link: https://ngdc.cncb.ac.cn/omix/release/OMIX002504. All data generated or analysed during the study are included in the manuscript and figure supplements. Source data files have been provided for Figures 1–4 as well as all the figure supplements. Raw gel data for Figure 4 and Figure 2—figure supplement 1 was uploaded as source data files corresponding to the figures.

The following dataset was generated:

| Author(s) | Year | Dataset title | Dataset URL | Database and Identifier |
|-----------|------|---------------|-------------|-------------------------|
| Liu C | 2022 | Expression matrix of BT474 cells treated with different durgs and their combination | https://ngdc.cncb.ac.cn/omix/release/OMIX002504 | GSA database, OMIX002504 |

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
