## [Editor Report]

This study presents a valuable finding on the combination use of pyrotinib, tamoxifen and dalpiciclib against HER2+/HR+ breast cancer. The evidence supporting the claims of the authors is solid. The work will be of interest to medical biologists or clinicians working on breast cancer.

---

## [Decision Letter]

**Decision letter after peer review:**

[Editors’ note: the authors submitted for reconsideration following the decision after peer review. What follows is the decision letter after the first round of review.]

Thank you for submitting the paper "Dalpiciclib and Pyrotinib Exert Synergistic Antitumor Effects in Triple Positive Breast Cancer" for consideration by *eLife*. Your article has been reviewed by 2 peer reviewers, one of whom is a member of our Board of Reviewing Editors, and the evaluation has been overseen by a Reviewing Editor and a Senior Editor. The reviewers have opted to remain anonymous.

Comments to the Authors:

We are sorry to say that, after consultation with the reviewers, we have decided that this work will not be considered further for publication by *eLife*. However, if the authors are able to fully and comprehensively address all review concerns, we are open to examine a new submission, without guaranteeing acceptance.

Generally the reviewers agreed they had to really dig into the trial contexts of the different parts of clinical investigations presented by googling, to find out what the trials are and what are the actual drugs used in the trials cited.

In brief, it is very unclear which clinical cohorts have been used in which figures (and there is a lack of detailed trial description or CONSORT Flow Diagram for the samples used) and it was not clear whether there are overlaps of the cohorts. In addition, some of the neoadjuvant cohorts (aside from the small number of patients in pyrotinib study), must have been treated with trastuzumab + pertuzumab. These would have a very different effects compared to irreversible TKIs such as pyrotinib.

In addition, the authors have not supported their preclinical studies using the same anti-HER2 monoclonal antibodies used in the clinical cohorts cited in various figures. The reviewers noted that clinical trial (NCT04486911) used letrozole instead of tamoxifen but the authors didn't show any data using letrozole in cell lines there would not be comparability between the preclinical and clinical parts of this manuscript.

It was felt the revision of all these weaknesses would be beyond the scope and time that would normally be allowed.

*Reviewer #1 (Recommendations for the authors):*

From reading the introduction section and its references, the premise (and hence the hypothesis to address) of this paper is that ER activity may underpin part of the SECONDARY/acquired resistance to anti-HER2 inhibitors. There has certainly been previous preclinical literature support for this hypothesis (Wang, Morrison et al., 2011).

Clinical trials have been conducted on the basis of the preclinical literature but these tend to be early stage studies (e.g. phase 1 study of Palbociclib and T-DM1 in HER2+ metastatic/advanced breast cancer patients who progress subsequent to Trastuzumab (Haley, Batra et al., 2021)) so clinically, this preclinical hypothesis has certainly not been substantiated yet.

Nor do we understand the precise mechanisms of the potential CDK-HER2 crosstalk in these patients. Notably one should differentiate this hypothesis from clinical trials which are primarily designed to address primary resistance (using ER-HER2 combined targeting) in early breast cancer patients (Brandao, Maurer et al., 2020, Harbeck, Gluz et al., 2017).

The authors succeeded in demonstrating that pyrotinib (anti-HER2 TKI) combined with dalpiciclib (CDK4/6 inhibitor) showed better efficacy than pyrotinib combined with tamoxifen. Mechanistically, the authors also succeeded in showing pyrotinib induced ER nuclear translocation in the human triple-positive (ER+/PR+/HER2+, TPBC) breast cancer cell line BT474, a phenomenon which could be partially reversed by the addition of dalpiciclib, rather than tamoxifen. The weakness of the followup experiments to these findings, which were designed to further investigate whether the expression of HER2 (using HER2 overexpression plasmids) could affect the distribution of ER, lies in the non-physiological overexpression of HER2 in ER+ MCF7 cells.

The authors also claimed they showed in Figure 2d, in their ongoing clinical trial (NCT04486911), the nuclear ER expression levels of patients did not show significant elevations after the HER2-targeted therapy. However a major weakness exists as NCT04486911 is a single arm study of Pyrotinib Maleate, CDK4/6 Inhibitor (dalpiciclib) and Letrozole in Combination for Stage II-III TPBC: a Phase II Trial.

As Figures 2c and 2d (third column labelled as CDKi+antiHER2) were not derived from the same trial (by definition as 2d's findings pertained to a single arm study), the claim that "these findings verified that the ER receptor may have shifted to the nucleus after anti-HER2 therapy, which could be reversed with the introduction of a CDK4/6 inhibitor" is not valid.

In BT474 cells, the authors concluded that after the introduction of dalpiciclib, the activation of mTOR was partially inhibited, which relieved the negative feedback on the HER2 pathway, as evidenced by the slight increase in the HER2 and pAKT, which maintained the sensitivity of the HER2 pathway to pyrotinib (Figure 4a). If this conclusion was drawn by comparing lanes 1 & 4 in Figure 4a (pTOR vs total TOR, compared also with total or phosphor- HER2 changes), the changes on the blots are not convincing to draw this conclusion.

Finally, the authors have succeeded in demonstrating that TPBC patients with positive CALML5 (being one of the three genes that were found to overlap between the upregulated genes treated with pyrotinib and the genes belonging to the estrogen signaling pathway (Figure 3G)), may benefit from the addition of CDK4/6 inhibitors to anti-HER2 antibody in neoadjuvant therapy (Figure 4c).

Brandao M, Maurer C, Ziegelmann PK, Ponde NF, Ferreira A, Martel S, Piccart M, de Azambuja E, Debiasi M, Lambertini M (2020) Endocrine therapy-based treatments in hormone receptor-positive/HER2-negative advanced breast cancer: systematic review and network meta-analysis. ESMO Open 5

Ding J, Kuang P (2021) Regulation of ERalpha Stability and Estrogen Signaling in Breast Cancer by HOIL-1. Front Oncol 11: 664689

Haley B, Batra K, Sahoo S, Froehlich T, Klemow D, Unni N, Ahn C, Rodriguez M, Hullings M, Frankel AE (2021) A Phase I/Ib Trial of PD 0332991 (Palbociclib) and T-DM1 in HER2-Positive Advanced Breast Cancer After Trastuzumab and Taxane Therapy. Clin Breast Cancer

Harbeck N, Gluz O, Christgen M, Kates RE, Braun M, Kuemmel S, Schumacher C, Potenberg J, Kraemer S, Kleine-Tebbe A, Augustin D, Aktas B, Forstbauer H, Tio J, von Schumann R, Liedtke C, Grischke EM, Schumacher J, Wuerstlein R, Kreipe HH et al. (2017) De-Escalation Strategies in Human Epidermal Growth Factor Receptor 2 (HER2)-Positive Early Breast Cancer (BC): Final Analysis of the West German Study Group Adjuvant Dynamic Marker-Adjusted Personalized Therapy Trial Optimizing Risk Assessment and Therapy Response Prediction in Early BC HER2- and Hormone Receptor-Positive Phase II Randomized Trial-Efficacy, Safety, and Predictive Markers for 12 Weeks of Neoadjuvant Trastuzumab Emtansine With or Without Endocrine Therapy (ET) Versus Trastuzumab Plus ET. J Clin Oncol 35: 3046-3054

Wang YC, Morrison G, Gillihan R, Guo J, Ward RM, Fu X, Botero MF, Healy NA, Hilsenbeck SG, Phillips GL, Chamness GC, Rimawi MF, Osborne CK, Schiff R (2011) Different mechanisms for resistance to trastuzumab versus lapatinib in HER2-positive breast cancers--role of estrogen receptor and HER2 reactivation. Breast Cancer Res 13: R121

Figure 2b shows that the expression of HER2 (using HER2 overexpression plasmids) could affect the distribution of ER, but this was demonstrated via non-physiological overexpression of HER2 in ER+ MCF7 cells. In addition to the results in Figure 2B, experiments should be done with knockout/knockdown with rescue by reintroducing physiological levels of HER2.

In Figure2—figure supplement 1a-b, the authors showed although the nuclear ER levels increased considerably after pyrotinib, the total expression of ER was reduced. The authors should show whether ER ubiquitination, a known mechanism to regulate ERα stability (Ding & Kuang, 2021), was increased.

As pointed out, data used to support the conclusion drawn by comparing lanes 1 & 4 in Figure 4a (pTOR vs total TOR, compared also with total or phosphor- HER2 changes), are not conclusive. Statistical differences of these purported changes in multiple independent biological repeats should be presented.

The authors also claimed they showed in Figure 2d, in their ongoing clinical trial (NCT04486911), the nuclear ER expression levels of patients did not show significant elevations after the HER2-targeted therapy. However they should clarify the study design of NCT04486911 which is a single-center, single-arm, open-label trial from what I can find – https://clinicaltrials.gov/ct2/show/NCT04486911. Otherwise, Figures 2c and 2d (third column labelled as CDK1+antiHER2) could not have been derived from the same trial and the claim that "these findings verified that the ER receptor may have shifted to the nucleus after anti-HER2 therapy, which could be reversed with the introduction of a CDK4/6 inhibitor" would not be valid. The authors should address this.

The authors should make clear in the results and the figure legends the exact detail of the clinical studies e.g. WRT "patients with positive CALML5 may benefit from the addition of CDK4/6 inhibitors to anti-HER2 antibody in neoadjuvant therapy (Figure 4c)", it is not clear what anti-HER2 was used and in which trial context.

*Reviewer #2 (Recommendations for the authors):*

The authors compared the effects of dalpiciclib, pyrotinib and tamoxifen or their combination in in triple-positive breast cancer cells. They showed synergistic effects of dalpiciclib and pyrotinib as well as pyrotinib and tamoxifen but not with tamoxifen and dalpiciclib although the greatest efficacy was seen with the triple combination. This study has also assessed potential predictive biomarker, CALML5 to CDK4/6 inhibitor in combination with anti-HER2.

Strengths

This paper has shown the promising anti-tumour effect of dalpiciclib and pyrotinib +/- tamoxifen. The study proposed the mechanisms of why dalpiciclib + pyrotinib was more effective than pyrotinib with tamoxifen since pyrotinib induced ER nuclear translocation, which could be partially reversed by the addition of dalpiciclib, rather than tamoxifen. The study utilizes human tissues, which strengthen the data.

Weakness

The promising combination of pyrotinib + CDk4/6 inhibitor has already been previously reported so the combination of irreversible TKI and CDK4/6 inhibitor combination is not novel. The paper referred to anti-HER2 treatments in neoadjuvant/adjuvant setting and this is likely to be trastuzumab and pertuzumab and would have different effects to irreversible TKI like pyrotinib. In addition, one of the clinical trial samples were from patients treated with letrozole but the authors have not supported the preclinical data using letrozole. The effect of tamoxifen or letrozole in combination with protinib and dalpaciclib may be different. The authors did not show the combination in xenograft and/or patient-derived organoid models, which would strengthen the data.

Overall, the authors' claims and conclusions are justified by their data but further work is required before the findings could be translated into clinic.

Figure 1: P+D and P+T effective and synergistic. T+D not effective or synergistic. In the combination treatment group (Figure 1C) there is no control group like DMSO. Use of t-test instead of anova in Figure 1C, not be appropriate as it causes bias in the selection of groups and does not account for multiple-testing errors. Could also use control (vehicle group). In addition, no error bars are shown in each point despite the figure legend states that data was presented as mean (plus minus) SEMs in Figure 1a-b.

Figure 1d – is this adjuvant or neoadjuvant – The Figure 1d states adjuvant but the table 1 and 2 state neoadjuvant, The cohorts of patients are very confusing. Is the 177 patients in Figure 1D the same as those 172 patients in Table 1? I am not sure whether Table 1, Table 2 or Figure 1d are from the same cohorts with overlaps or are there complete different cohorts. If Figure 1d is from different cohort, patients' demographic and characteristics need to be shown. If they are the same cohorts, the manuscript needs to be clearer.

I think Figure 2c is misleading and doesn't necessarily support the data from cell lines. This is because most the anti-HER2 and chemotherapy given to patients would be different from those in the in vitro experiments. The authors have now shown the effect of trastuzumab and pertuzumab +/- chemo on ER nuclear translocation in cell line

The author stated their ongoing clinical trial (NCT04486911) and that the nuclear ER expression levels of patients did not show significant elevations after the HER2-targeted therapy combined with dalpiciclib (Figure 2d). However, in this study, the hormone treatment was different, letrozole instead of tamoxifen. The authors didn't show any data using letrozole in cell lines.

It is important for the authors to state the drugs used in NCT04486911 – otherwise we have to google to find out. The authors should have stated that dalpiciclib is SHR6390 used in NCT04486911.

In both table 1 and 2, the HER2 status contained both 2+ and 3+. I assumed that the HER2 2+ was FISH positive and if so this needs to be stated.

Figure 2B. Not sure what does NC stand for. This should be stated.

Figure 3: CALML5 is indicated to be a predictive biomarker for responsiveness to anti-her2 therapy containing CDK4/6 inhibitor. The authors stated that they wanted to further explore the synergistic mechanisms of the addition of CDK4/6 inhibitor treatment in HER2+/HR+ breast cancer. They first analyzed the gene expression profiles of the breast tumor cells treated with pyrotinib via RNA-seq. They then compared gene expression between pyrotinib, tamoxifen and dalpaciclib and pyrotinib and tamoxifen. I am surprised that they did not compare the gene expression between pyrotinib + dalpiciclib with pyrotinib alone to see the effect of adding dalpaciclib to anti-HER2 palbociclib as they wanted to explore the synergistic mechanisms of adding CDK4/6 inhibitor to anti-Her2.

Could the authors define CALML5-positive cells – is it any staining or just strong staining and % of positive cells?

In table 2 – it states that the patient received chemotherapy + anti-HER2 – which anti-Her2 therapies? If they are trastuzumab and pertuzumab, these would have induced very different effect compared to irreversible TKI.

Figure 4: 4a the spelling for pyrotinib is wrong in the figure. There is no control treatment group using DMSO. Phosphorylation status of CDK4/6 could be useful to look at because its activity is regulated by Cyclin D and p27 and phosphorylation of the threonine 172 residue (pThr172) is the rate limiting step in CDK4 activation. Therefore, tumour cells lacking Thr172 phosphorylation would progress in cell cycle independent of CDK4 pathway.

[Editors’ note: further revisions were suggested prior to acceptance, as described below.]

Thank you for submitting your article "Dalpiciclib Partially Abrogates ER Signaling Activation Induced by Pyrotinib In HER2 + HR + Breast Cancer" for consideration by *eLife*. Your article has been reviewed by 3 peer reviewers, and the evaluation has been overseen by a Reviewing Editor and Mone Zaidi as the Senior Editor. The following individual involved in the review of your submission has agreed to reveal their identity: Huihui Li (Reviewer #3).

Essential revisions:

1) Language inconsistency and grammar mistakes throughout the manuscript must be fixed. This is mandatory.

2) Change "CDK4/6 inhibitor" into "dalpiciclib" in line 221.

3) In Table 1 and Table 2, the reasons for dividing the demographic data into subgroups should be specified in the Material and Method session.

*Reviewer #1 (Recommendations for the authors):*

The manuscript discussed the combination use of pyrotinib, tamoxifen, and dalpiciclib against HER2+/HR+ breast cancer cells. Through a series of in vitro drug sensitivity studies and in vivo drug susceptibility studies, the authors revealed that pyrotinib combined with dalpiciclib exhibits better therapeutic efficacy than the combination use of pyrotinib with tamoxifen. Moreover, the authors found that CALML5 may serve as a biomarker in the treatment of HER2+/HR+ breast cancer.

The authors provide solid evidence for the following:

1. The combination use of pyrotinib with dalpiciclib exhibits better therapeutic efficacy than the combination use of pyrotinib with tamoxifen.

2. Nuclear ER distribution is increased upon anti-HER2 therapy and could be partially abrogated by the treatment of dalpiciclib.

3. CALML5 may serve as a putative risk biomarker in the treatment of HER2+/HR+ breast cancer.

The manuscript has significant strengths and several weaknesses. The strengths include the identification of the novel role of dalpiciclib in the treatment of HER2+/HR+ breast cancer. Moreover, the authors provide solid evidence that the combined use of dalpiciclib with pyrotinib significantly decreased the total and nuclear expression of ER. The main weakness of the manuscript is that the manuscript is difficult to read due to language inconsistency. In addition, some figure captions and figure legends should be carefully amended.

Below are some specific points that I believe would certainly strengthen the presentation,

1. Language inconsistency and grammar mistakes throughout the manuscript must be fixed. This is mandatory.

2. For instance, on page 5, lines 90-92, "Pyrotinib combined with dalpiciclib shows better cytotoxic efficacy than when combined with tamoxifen", what does this mean exactly? Please specify.

3. Page 2, line 18, "…remain further investigation"? should be "remained elusive" or "warrants further investigation".

4. Page 2, line 24, 'selected out' should be 'identified'; 'tested' should be 'ascertained' or 'evaluated'.

5. Page 2, line 33, 'overcome this', overcome what? Please specify.

6. "western blot", 'Western' should always be capitalized.

7. IC50, '50' should be subscript.

8. Page 6, line 115, the sentence should be fixed.

9. Page 7, line 151, "shifted"? should be "relocated".

10. Page 7, line 161, the sentence should be fixed.

11. Page 11, line 241, the sentence should be fixed.

12. Page 12, line 264, the sentence should be fixed.

13. Page 12, line 274, 'displayed' should be 'identified'.

14. Page 13, line 290, 'determine' should be 'assess'.

15. Figure 1a, the captions of the vertical axis were missing for two panels.

16. Figure 1b and 1c, 'Cell Viability' should be 'Cell viability'.

17. Figure 2, 'cell ratio' should be 'Cell ratio'.

18. Figure 4, the font size of the captions should be adjusted to the same.

19. Figure 1, Supplement 1, b, the last panel should be adjusted to a circle. The vertical axis, 'colonies formated' should be 'Colony formation'.

20. Figure 2, Supplement 1, c, the vertical axis, 'cell ratio' should be 'Cell ratio'. These figure captions should be kept consistent in the paper.

*Reviewer #2 (Recommendations for the authors):*

The authors performed preclinical studies to investigate the underlying mechanism of how the combination of pyrotinib, letrozole and dalpiciclib achieved satisfactory clinical outcomes in the MUKDEN 01 clinical trial (NCT04486911). Mechanistically, using anti-HER2 drugs such as pyrotinib and trastuzumab could degrade HER2 and facilitate the nuclear transportation of ER in HER2+HR+ breast cancer, which enhanced the function of ER signaling pathway. The introduction of dalpiciclib partially abrogated the nuclear transportation of ER and exerted its canonical function as cell cycle blockers, which led to the optimal cytotoxicity effect in treating HER2+HR+ breast cancer. Furthermore, using mRNA-seq analysis and in vivo drug susceptibility test, the authors succeeded in identifying CALML5 as a novel risk factor in the treatment of HER2+HR+ breast cancer.

1. The catalogue number of the antibodies used in this study shall be added in the section of Chemicals and antibodies.

2. I suggest the change of "CDK4/6 inhibitor" into "dalpiciclib" in line 221, because only dalpiciclib was used in this study and we were unaware of if other CDK4/6 inhibitors could affect the nuclear transportation of ER.

*Reviewer #3 (Recommendations for the authors):*

In this research, the authors explore a novel mechanism of CDK4/6 inhibitor dalpiciclib in HER2+HR+ breast cancers, in which dalpiciclib could reverse the process of ER intra-nuclear transportation upon HER2 degradation. The conclusions are significant to gain insight into the biological behavior of TPBC and provided a conceptual basis for the ideal efficacy in the published clinical trial. The findings are supported by supplemented in vivo assay and transcriptomic analysis.

1. In some parts of the manuscript, the author interchanged the expression of "breast cancer" and "breast tumor", I suggest sticking to one expression throughout the whole text, which may improve the concordance of the manuscript.

2. In Table 1 and Table 2, the reasons for dividing the demographic data into subgroups should be specified in the Material and Method session.

3. In Table 1 and Table 2, the first line where different treatment groups were listed should be adjusted for new line management.

---

## [Author Response]

[Editors’ note: the authors resubmitted a revised version of the paper for consideration. What follows is the authors’ response to the first round of review.]

Comments to the Authors:We are sorry to say that, after consultation with the reviewers, we have decided that this work will not be considered further for publication by eLife. However, if the authors are able to fully and comprehensively address all review concerns, we are open to examine a new submission, without guaranteeing acceptance.Generally the reviewers agreed they had to really dig into the trial contexts of the different parts of clinical investigations presented by googling, to find out what the trials are and what are the actual drugs used in the trials cited.In brief, it is very unclear which clinical cohorts have been used in which figures (and there is a lack of detailed trial description or CONSORT Flow Diagram for the samples used) and it was not clear whether there are overlaps of the cohorts. In addition, some of the neoadjuvant cohorts (aside from the small number of patients in pyrotinib study), must have been treated with trastuzumab + pertuzumab. These would have a very different effects compared to irreversible TKIs such as pyrotinib.In addition, the authors have not supported their preclinical studies using the same anti-HER2 monoclonal antibodies used in the clinical cohorts cited in various figures. The reviewers noted that clinical trial (NCT04486911) used letrozole instead of tamoxifen but the authors didn't show any data using letrozole in cell lines there would not be comparability between the preclinical and clinical parts of this manuscript.

Thanks for your invaluable input into improving our manuscript. We’ve added up the detailed information of the clinical cohorts we used in the results part (line 138-146, 202-211), methods part (line 278-284) as well as the figure legends. We also made new tables to describe the demographic information of the clinical samples we analyzed (Table1 and 2). As for the difference between trastuzumab and pyrotinib, we treated our cells using trastuzumab and combined with the other drugs and received similar results in affecting ER distribution in BT474 cells as pyrotinib did (Figure 2—figure supplement 1 c).

We admit that using tamoxifen instead of letrozole may not be quite appropriate in the in vitro studies. However, as an aromatase inhibitor, letrozole could only exert functions when treating cells which overexpressed aromatase (*Banerjee et al.*, *2010*). The over expression of aromatase in BT474 cells lines may cause other alterations in the cell nature properties and may not simulate the HER2^+^/HR^+^ breast cancer very well. Hence, we used the tamoxifen which could directly inhibit ER as the endocrine therapy.

Reviewer #1 (Recommendations for the authors):From reading the introduction section and its references, the premise (and hence the hypothesis to address) of this paper is that ER activity may underpin part of the SECONDARY/acquired resistance to anti-HER2 inhibitors. There has certainly been previous preclinical literature support for this hypothesis (Wang, Morrison et al., 2011).Clinical trials have been conducted on the basis of the preclinical literature but these tend to be early stage studies (e.g. phase 1 study of Palbociclib and T-DM1 in HER2+ metastatic/advanced breast cancer patients who progress subsequent to Trastuzumab (Haley, Batra et al., 2021)) so clinically, this preclinical hypothesis has certainly not been substantiated yet.Nor do we understand the precise mechanisms of the potential CDK-HER2 crosstalk in these patients. Notably one should differentiate this hypothesis from clinical trials which are primarily designed to address primary resistance (using ER-HER2 combined targeting) in early breast cancer patients (Brandao, Maurer et al., 2020, Harbeck, Gluz et al., 2017).

We agree with the reviewer’s idea about that ER activity may underpin part of the SECONDARY/acquired resistance to anti-HER2 inhibitors and this is one of the most important views in our article. In the preclinical literature support for this hypothesis by(*Wang et al.*, *2011*), the solution to acquired resistance to anti-HER2 inhibitors was emphasized on the blockade of ER. However, in our study, we found that despite the combination of anti-HER2 therapy and blockade of ER showed synergistic effect, it was still not efficacy enough and the introduction of CDK4/6 inhibitor might be a better way to solve the re-activation of ER (shift into nucleus) due to anti-HER2 therapy. Moreover, through mRNA-seq analysis and in vivo drug sensitivity tests (Figure 4 e and f)，we revealed CALML5 as a potential risk factor in the treatment of HER2^+^HR^+^ breast cancer and the introduction of dalpiciclib might overcome this.

The authors succeeded in demonstrating that pyrotinib (anti-HER2 TKI) combined with dalpiciclib (CDK4/6 inhibitor) showed better efficacy than pyrotinib combined with tamoxifen. Mechanistically, the authors also succeeded in showing pyrotinib induced ER nuclear translocation in the human triple-positive (ER+/PR+/HER2+, TPBC) breast cancer cell line BT474, a phenomenon which could be partially reversed by the addition of dalpiciclib, rather than tamoxifen. The weakness of the followup experiments to these findings, which were designed to further investigate whether the expression of HER2 (using HER2 overexpression plasmids) could affect the distribution of ER, lies in the non-physiological overexpression of HER2 in ER+ MCF7 cells.

Thanks for your comments and suggestions in our findings. As for the investigation of whether expression of HER2 could affect the distribution of ER, the published article “Human Epidermal Growth Factor Receptor 2 Status Modulates Subcellular Localization of and Interaction with Estrogen Receptor in Breast Cancer Cells”(*Yang et al.*, *2004*) had sufficiently demonstrated the relationship between the distribution of ER and the expression of HER2 and we’ve cited this article (line 121-125) to prove our ideas and removed the part of overexpression HER-2 plasmid in ER^+^ MCF7 cells (Figure 2b in the previous version) to avoid confusion. Besides, through our experimental results, we found that targeting HER2 using TKIs or trastuzumab could affect the distribution of ER in BT474 cells (Figure 2a and Figure 2—figure supplement 1c).

The authors also claimed they showed in Figure 2d, in their ongoing clinical trial (NCT04486911), the nuclear ER expression levels of patients did not show significant elevations after the HER2-targeted therapy. However a major weakness exists as NCT04486911 is a single arm study of Pyrotinib Maleate, CDK4/6 Inhibitor (dalpiciclib) and Letrozole in Combination for Stage II-III TPBC: a Phase II Trial.As Figures 2c and 2d (third column labelled as CDKi+antiHER2) were not derived from the same trial (by definition as 2d's findings pertained to a single arm study), the claim that "these findings verified that the ER receptor may have shifted to the nucleus after anti-HER2 therapy, which could be reversed with the introduction of a CDK4/6 inhibitor" is not valid.

Thanks for your pointing out about the weakness about our clinical evidence. Indeed, 3 kinds of neoadjuvant therapy were enrolled in this part to evaluate if anti-HER therapy could lead to ER nuclear shift in clinical practice. The therapies were as follow: trastuzumab+chemotherapy(docetaxel+carboplatin); chemotherapy (docetaxel+carboplatin); pyrotinib+dalpiciclib+letrozole which is our clinical trial NCT04486911 (line 138-146, 202-211, 278-284). The demographic information of these patients was described in Table 1. Despite there may be some difference in the working mechanism of trastuzumab and pyrotinib in targeting HER2, the affecting of ER distribution was both observed in clinical specimens (Figure 2b) and cell lines (Figure 2—figure supplement 1c). Hence, we think these evidences could still partially prove our idea that the usage of anti-HER2 therapy could enhance the nuclear distribution of ER in HER2^+^HR^+^breast cancer.

In BT474 cells, the authors concluded that after the introduction of dalpiciclib, the activation of mTOR was partially inhibited, which relieved the negative feedback on the HER2 pathway, as evidenced by the slight increase in the HER2 and pAKT, which maintained the sensitivity of the HER2 pathway to pyrotinib (Figure 4a). If this conclusion was drawn by comparing lanes 1 & 4 in Figure 4a (pTOR vs total TOR, compared also with total or phosphor- HER2 changes), the changes on the blots are not convincing to draw this conclusion.

Thanks for your suggestion, the description of this part was a little bit overstated and we’ve checked out manuscript and removed such descriptions.

Finally, the authors have succeeded in demonstrating that TPBC patients with positive CALML5 (being one of the three genes that were found to overlap between the upregulated genes treated with pyrotinib and the genes belonging to the estrogen signaling pathway (Figure 3G)), may benefit from the addition of CDK4/6 inhibitors to anti-HER2 antibody in neoadjuvant therapy (Figure 4c).

Thanks for your comments. Besides, we added in vivo study to investigate the function of CALML5 in this version (line 213-218). The existence of CALML5 could lead to relative worse response to pyrotinib+tamoxifen and requires the introduction of dalpiciclib (Figure 4e and f).

Figure 2b shows that the expression of HER2 (using HER2 overexpression plasmids) could affect the distribution of ER, but this was demonstrated via non-physiological overexpression of HER2 in ER+ MCF7 cells. In addition to the results in Figure 2B, experiments should be done with knockout/knockdown with rescue by reintroducing physiological levels of HER2.

Thanks for your suggestions. Actually, our findings emphasized on the usage of anti-HER2 therapy to disturb the location of ER. As for the investigation of whether expression of HER2 could affect the distribution of ER, the published article “Human Epidermal Growth Factor Receptor 2 Status Modulates Subcellular Localization of and Interaction with Estrogen Receptor in Breast Cancer Cells” (*Yang et al.*, *2004*) had sufficiently demonstrated the relationship between the distribution of ER and the expression of HER2. Hence, we’ve cited this article to prove our ideas (line 121-124) that targeting HER2 using TKIs or trastuzumab could affect the distribution of ER (Figure 2a and Figure 2—figure supplement 1c) and removed the part of overexpressing HER2 plasmids in MCF7 cells (Figure 2b in the previous version).

In Figure 2—figure supplement 1a-b, the authors showed although the nuclear ER levels increased considerably after pyrotinib, the total expression of ER was reduced. The authors should show whether ER ubiquitination, a known mechanism to regulate ERα stability (Ding & Kuang, 2021), was increased.

Thanks for your suggestion, to investigate the status of ER ubiquitination in BT474 cells treated with different drugs, we performed ER ubiquitination in Figure 2-supplement 1 d. We found that the introduction of dalpiciclib significantly increased the ubiquitination of ER (line132-135) and this was consistent with the findings of ER expression in Figure 2-supplement 1 a and b.

As pointed out, data used to support the conclusion drawn by comparing lanes 1 & 4 in Figure 4a (pTOR vs total TOR, compared also with total or phosphor- HER2 changes), are not conclusive. Statistical differences of these purported changes in multiple independent biological repeats should be presented.

Thanks for your suggestion, we found the description of this part was a little bit overstated and we’ve checked out manuscript and changed such descriptions about the changes in pTOR vs total TOR (line 192-194).

The authors also claimed they showed in Figure 2d, in their ongoing clinical trial (NCT04486911), the nuclear ER expression levels of patients did not show significant elevations after the HER2-targeted therapy. However they should clarify the study design of NCT04486911 which is a single-center, single-arm, open-label trial from what I can find – https://clinicaltrials.gov/ct2/show/NCT04486911. Otherwise, Figures 2c and 2d (third column labelled as CDK1+antiHER2) could not have been derived from the same trial and the claim that "these findings verified that the ER receptor may have shifted to the nucleus after anti-HER2 therapy, which could be reversed with the introduction of a CDK4/6 inhibitor" would not be valid. The authors should address this.

Thanks for your suggestion, we should have clarified about the clinical trial. Indeed, 3 kinds of neoadjuvant therapy were enrolled in this part to elevate if anti-HER2 therapy could lead to ER nuclear shift in clinical practice. The therapies were as follow: trastuzumab+chemotherapy(docetaxel+carboplatin); chemotherapy (docetaxel+carboplatin); pyrotinib+dalpiciclib+letrozole which is our clinical trial NCT04486911 (line 138-146, 202-211, 278-284). The demographic information of these patients was described in Table 1. Despite there may be some difference in the working mechanism of trastuzumab and pyrotinib in targeting HER2, the affecting of ER distribution was both observed in clinical specimens (Figure 2b) and cell lines (Figure 2—figure supplement 1c). Hence, we think these evidences could still partially prove our idea that the usage of anti-HER2 therapy could enhance the nuclear distribution of ER in HER2^+^HR^+^breast cancer.

The authors should make clear in the results and the figure legends the exact detail of the clinical studies e.g. WRT "patients with positive CALML5 may benefit from the addition of CDK4/6 inhibitors to anti-HER2 antibody in neoadjuvant therapy (Figure 4c)", it is not clear what anti-HER2 was used and in which trial context.

Thanks for your suggestion. We’ve labeled the exact detail of drug use in the clinical studies in results part (line 138-146, 202-211), methods part (line 278-284) as well as figure legends. In this part, we mean that patients with positive CALML5 may benefit from dalpiciclib when treated by trastuzumab or TKIs (line 265-267).

Reviewer #2 (Recommendations for the authors):The authors compared the effects of dalpiciclib, pyrotinib and tamoxifen or their combination in in triple-positive breast cancer cells. They showed synergistic effects of dalpiciclib and pyrotinib as well as pyrotinib and tamoxifen but not with tamoxifen and dalpiciclib although the greatest efficacy was seen with the triple combination. This study has also assessed potential predictive biomarker, CALML5 to CDK4/6 inhibitor in combination with anti-HER2.StrengthsThis paper has shown the promising anti-tumour effect of dalpiciclib and pyrotinib +/- tamoxifen. The study proposed the mechanisms of why dalpiciclib + pyrotinib was more effective than pyrotinib with tamoxifen since pyrotinib induced ER nuclear translocation, which could be partially reversed by the addition of dalpiciclib, rather than tamoxifen. The study utilizes human tissues, which strengthen the data.WeaknessThe promising combination of pyrotinib + CDk4/6 inhibitor has already been previously reported so the combination of irreversible TKI and CDK4/6 inhibitor combination is not novel. The paper referred to anti-HER2 treatments in neoadjuvant/adjuvant setting and this is likely to be trastuzumab and pertuzumab and would have different effects to irreversible TKI like pyrotinib. In addition, one of the clinical trial samples were from patients treated with letrozole but the authors have not supported the preclinical data using letrozole. The effect of tamoxifen or letrozole in combination with protinib and dalpaciclib may be different. The authors did not show the combination in xenograft and/or patient-derived organoid models, which would strengthen the data.Overall, the authors' claims and conclusions are justified by their data but further work is required before the findings could be translated into clinic.

Thanks for the comments. We agree with the idea that the effect of tamoxifen or letrozole in combination with pyrotinib and dalpiciclib may be different. However, as an aromatase inhibitor, letrozole could functionally inhibit the generation of estrogen in human body and subsequently dysfunction ER. In in vitro conditions, letrozole could only exert functions when treating cells which overexpressed aromatase (Banerjee, 2010 #73). The over expression of aromatase in BT474 cells lines may cause other alterations in the cell properties and may not simulate the HER2^+^/HR^+^ breast cancer very well. Hence, we used the tamoxifen which could directly inhibit ER as the endocrine therapy in preclinical studies.

As for in vivo xenograft studies, we added in vivo xenografts to investigate the function of CALML5 in this part (line 213-221). CALML5 could serve as a potential risk factor in the treatment of HER2^+^HR^+^ breast cancer while the introduction of dalpiciclib might overcome this (Figure 4 e and f).

Figure 1: P+D and P+T effective and synergistic. T+D not effective or synergistic. In the combination treatment group (Figure 1C) there is no control group like DMSO. Use of t-test instead of anova in Figure 1C, not be appropriate as it causes bias in the selection of groups and does not account for multiple-testing errors. Could also use control (vehicle group). In addition, no error bars are shown in each point despite the figure legend states that data was presented as mean (plus minus) SEMs in Figure 1a-b.

Thanks for your suggestions. As for DMSO control group in Figure 1 c, actually we performed this group but didn’t show it in the previous version and we’ve displayed this group in Figure 1 c this time. The calculation in Figure 1 c was performed using anova test among different groups and we ‘ve announced this in the methods part. In Figure 1 a-b, we’ve changed the error bars into SDs, so that all the error bars could be seen clearly.

Figure 1d – is this adjuvant or neoadjuvant – The Figure 1d states adjuvant but the table 1 and 2 state neoadjuvant, The cohorts of patients are very confusing. Is the 177 patients in Figure 1D the same as those 172 patients in Table 1? I am not sure whether Table 1, Table 2 or Figure 1d are from the same cohorts with overlaps or are there complete different cohorts. If Figure 1d is from different cohort, patients' demographic and characteristics need to be shown. If they are the same cohorts, the manuscript needs to be clearer.

Thanks for your comments and sorry for the cause of this confusion. When we were revising this article, we realized that the data of Figure 1 d (previous version) was not quite relevant to our study and we removed this part in this version. As for the clinical samples we used in other parts, we’ve labeled them clearly in the result part (line 138-146, 202-211), method part (line 278-284) as well as the figure legends.

I think Figure 2c is misleading and doesn't necessarily support the data from cell lines. This is because most the anti-HER2 and chemotherapy given to patients would be different from those in the in vitro experiments. The authors have now shown the effect of trastuzumab and pertuzumab +/- chemo on ER nuclear translocation in cell line

We agree with your idea that the clinical use of anti-HER2 therapy given to patients would be different from in vitro studies. Followed your suggestion, we treated BT474 cell line using trastuzumab and found the similar result about ER translocation as pyrotinib (Figure 2—figure supplement 1 c). We believe that evidence from clinical samples could partially support our idea that the usage of anti-HER2 therapy could lead to ER translocation hence we kept Figure 2 b (Figure 2 c in previous version).

The author stated their ongoing clinical trial (NCT04486911) and that the nuclear ER expression levels of patients did not show significant elevations after the HER2-targeted therapy combined with dalpiciclib (Figure 2d). However, in this study, the hormone treatment was different, letrozole instead of tamoxifen. The authors didn't show any data using letrozole in cell lines.

Thanks for your comments and concern. We admit that using tamoxifen instead of letrozole may not be quite appropriate in the in vitro studies. However, as an aromatase inhibitor, letrozole could only exert functions when treating cells which overexpressed aromatase (Banerjee, 2010 #73). The over expression of aromatase in BT474 cells lines may cause other alterations in the cell nature properties and may not simulate the HER2^+^/HR^+^ breast cancer very well. Hence, we used the tamoxifen which could directly inhibit ER as the endocrine therapy in this study.

It is important for the authors to state the drugs used in NCT04486911 – otherwise we have to google to find out. The authors should have stated that dalpiciclib is SHR6390 used in NCT04486911.

Thanks for your suggestion and sorry for the cause of your confusion and inconvenience. We’ve stated the drugs used in NCT04486911 in the result part (line 138-146, 202-211) as well as the method part (line 278-284) this time. The drug dalpiciclib is SHR6390 used in NCT04486911.

In both table 1 and 2, the HER2 status contained both 2+ and 3+. I assumed that the HER2 2+ was FISH positive and if so this needs to be stated.

Thanks for your suggestion. The HER2 status labeled with 2+ were detected by FISH and we stated this in the methods part this time (line 386-387).

Figure 2B. Not sure what does NC stand for. This should be stated.

In the previous manuscript we submitted, NC here represented for negative control plasmids. We removed the part of overexpression HER2 plasmids in MCF7 cell line for the following reason. The published article “Human Epidermal Growth Factor Receptor 2 Status Modulates Subcellular Localization of and Interaction with Estrogen Receptor in Breast Cancer Cells” (*Yang et al.*, *2004*) had sufficiently demonstrated the relationship between the distribution of ER and the expression of HER2 and we’ve cited this article to prove our ideas that targeting HER2 using TKIs or Trastuzumab could affect the distribution of ER (Figure 2a and Figure 2—figure supplement 1c).

Figure 3: CALML5 is indicated to be a predictive biomarker for responsiveness to anti-her2 therapy containing CDK4/6 inhibitor. The authors stated that they wanted to further explore the synergistic mechanisms of the addition of CDK4/6 inhibitor treatment in HER2+/HR+ breast cancer. They first analyzed the gene expression profiles of the breast tumor cells treated with pyrotinib via RNA-seq. They then compared gene expression between pyrotinib, tamoxifen and dalpaciclib and pyrotinib and tamoxifen. I am surprised that they did not compare the gene expression between pyrotinib + dalpiciclib with pyrotinib alone to see the effect of adding dalpaciclib to anti-HER2 palbociclib as they wanted to explore the synergistic mechanisms of adding CDK4/6 inhibitor to anti-Her2.

Thanks for your comments. This is because that we found the usage of pyrotinib may activate the ER signaling pathway. Through analyzing the differential expressed genes between pyrotinib group and DMSO control group, we may find out potential markers which could reflect the ER signaling activation. Further on, to seek how dalpiciclib effectively inhibit the cell viability, we intersected the genes which was up-regulated by the introduction of pyrotinib and was down-regulated by dalpiciclib. As we believe this down stream genes was also controlled by ER signaling, we are afraid that the introduction of tamoxifen could partially block its change and make the change of the expression not significant, hence we intersected the up-regulated genes in pyrotinib treatment and down-regulated genes in dalpiciclib treatment(line174-185).

Could the authors define CALML5-positive cells – is it any staining or just strong staining and % of positive cells?

Thanks for your suggestion. Any staining of cancer cells was defined as CALML5 positive cells and the specimens which barely got any staining were defined as CALML5 negative cells in Figure 4c.

In table 2 – it states that the patient received chemotherapy + anti-HER2 – which anti-Her2 therapies? If they are trastuzumab and pertuzumab, these would have induced very different effect compared to irreversible TKI.

Thanks for your comments. The chemotherapy+anti-HER2 therapy referred to docetaxel+carboplatin+trastuzumab. We understand the concern about the difference between trastuzumab and pyrotinib, hence we performed the effect of trastuzumab on ER translocation in BT474 cells and found similar results as pyrotinib did (Figure 2—figure supplement 1c). Based on these evidences, we believe that the clinical samples could still offer some proof about the anti-HER2 therapy and ER translocation.

Figure 4: 4a the spelling for pyrotinib is wrong in the figure. There is no control treatment group using DMSO. Phosphorylation status of CDK4/6 could be useful to look at because its activity is regulated by Cyclin D and p27 and phosphorylation of the threonine 172 residue (pThr172) is the rate limiting step in CDK4 activation. Therefore, tumour cells lacking Thr172 phosphorylation would progress in cell cycle independent of CDK4 pathway.

Thanks for the suggestion and sorry for the spelling mistake. We’ve double check the manuscript and changed this mistake. Actually, the first lane in the western blot images was the DMSO treated control group and we’ve labeled it in this version. We performed the western blot analysis on pCDK4 (pThr172) and found BT474 had physiologically expression of pCDK4 (pThr172) (Figure 4 a, line 195-196).

References:

Banerjee, S, A'Hern, R, Detre, S, Littlewood-Evans, AJ, Evans, DB, Dowsett, M, and Martin, LA. 2010. Biological evidence for dual antiangiogenic-antiaromatase activity of the VEGFR inhibitor PTK787/ZK222584 in vivo. Clin Cancer Res 16, 4178-4187. DOI: https://doi.org/10.1158/1078-0432.CCR-10-0456, PMID:20682704.

Wang, YC, Morrison, G, Gillihan, R, Guo, J, Ward, RM, Fu, X, Botero, MF, Healy, NA, Hilsenbeck, SG, Phillips, GL, Chamness, GC, Rimawi, MF, Osborne, CK, and Schiff, R. 2011. Different mechanisms for resistance to trastuzumab versus lapatinib in HER2-positive breast cancers--role of estrogen receptor and HER2 reactivation. Breast Cancer Res 13, R121. DOI: https://doi.org/10.1186/bcr3067, PMID:22123186.

Yang, Z, Barnes, CJ, and Kumar, R. 2004. Human epidermal growth factor receptor 2 status modulates subcellular localization of and interaction with estrogen receptor α in breast cancer cells. Clin Cancer Res 10, 3621-3628. DOI: https://doi.org/10.1158/1078-0432.CCR-0740-3, PMID:15173068.

[Editors’ note: what follows is the authors’ response to the second round of review.]

Essential revisions:1) Language inconsistency and grammar mistakes throughout the manuscript must be fixed. This is mandatory.

Thanks for your kindly suggestions to help us to improve this manuscript, we’ve improved the language expressions and fixed the grammar mistakes according to the reviewers’ suggestions.

2) Change "CDK4/6 inhibitor" into "dalpiciclib" in line 221.

Thanks for your suggestions and we’ve changed this in the revised manuscript.

3) In Table 1 and Table 2, the reasons for dividing the demographic data into subgroups should be specified in the Material and Method session.

Actually, Table 1 and Table 2 contained patients from the same clinical trial (patients who received chemotherapy+trastuzumab as neoadjuvant therapy and patients who received pyrotinib+dalpiciclib+letrozole as neoadjuvant therapy). In Table 1, patients who only received chemotherapy was not evaluated for the expression of CALML5 before receiving neoadjuvant therapy. To avoid confusions when mentioning the patients who were evaluated for the expression of CALML5 in the manuscript, we additionally made Table 2 to display the demographic characteristics of these patients.

Reviewer #1 (Recommendations for the authors):The manuscript discussed the combination use of pyrotinib, tamoxifen, and dalpiciclib against HER2+/HR+ breast cancer cells. Through a series of in vitro drug sensitivity studies and in vivo drug susceptibility studies, the authors revealed that pyrotinib combined with dalpiciclib exhibits better therapeutic efficacy than the combination use of pyrotinib with tamoxifen. Moreover, the authors found that CALML5 may serve as a biomarker in the treatment of HER2+/HR+ breast cancer.The authors provide solid evidence for the following:1. The combination use of pyrotinib with dalpiciclib exhibits better therapeutic efficacy than the combination use of pyrotinib with tamoxifen.2. Nuclear ER distribution is increased upon anti-HER2 therapy and could be partially abrogated by the treatment of dalpiciclib.3. CALML5 may serve as a putative risk biomarker in the treatment of HER2+/HR+ breast cancer.The manuscript has significant strengths and several weaknesses. The strengths include the identification of the novel role of dalpiciclib in the treatment of HER2+/HR+ breast cancer. Moreover, the authors provide solid evidence that the combined use of dalpiciclib with pyrotinib significantly decreased the total and nuclear expression of ER. The main weakness of the manuscript is that the manuscript is difficult to read due to language inconsistency. In addition, some figure captions and figure legends should be carefully amended.

Thanks for your comments on our manuscript. We feel sincerely sorry for the inconsistency of the manuscript due to poor language. We have improved our manuscript as well as the figures according to your valuable suggestions.

Below are some specific points that I believe would certainly strengthen the presentation,1. Language inconsistency and grammar mistakes throughout the manuscript must be fixed. This is mandatory.

Thanks for your valuable suggestions, we’ve fixed our language and grammar mistakes where applicable. Besides, we’ve improved our manuscript as well as the figures according to your kindly suggestions below.

2. For instance, on page 5, lines 90-92, "Pyrotinib combined with dalpiciclib shows better cytotoxic efficacy than when combined with tamoxifen", what does this mean exactly? Please specify.

Thanks for your comments and sorry for the confusion in our subtitles.

In this part, we mean that in the in vitro drug susceptibility tests, pyrotinib combined with dalpiciclib exerted better cytotoxicity on BT474 cells than pyrotinib combined with tamoxifen. This was justified by the in vitro drug susceptibility tests in Figure 1 c, where pyrotinib, dalpiciclib and tamoxifen was combined to each other at their IC_50_ and half IC_50_ concentration. To avoid further confusions and inconvenience in reading the manuscript, we’ve changed the subtitle into “Pyrotinib combined with dalpiciclib exerted stronger cytotoxic effect than pyrotinib combined with tamoxifen”.

3. Page 2, line 18, "…remain further investigation"? should be "remained elusive" or "warrants further investigation".

Thanks for your suggestion, we’ve changed this sentence into “the underlying molecular mechanism remained elusive” (line 18, line 244 and line 274).

4. Page 2, line 24, 'selected out' should be 'identified'; 'tested' should be 'ascertained' or 'evaluated'.

Thanks for your suggestions, we’ve improved the expression in this part according to your valuable suggestions (line 23, line 277).

5. Page 2, line 33, 'overcome this', overcome what? Please specify.

Thanks for your comments and sorry for the confusion in the manuscript. In this part, we mean that the introduction of dalpiciclib in the treatment of HER2^+^HR^+^ breast cancer could overcome the drug resistance to pyrotinib+tamoxifen due to CALML5 expression. We have made the expression more specified in this part as well as other parts in the manuscript (line 32, line 82-83, line 224-225, line 269-272 and line 279-280).

6. "western blot", 'Western' should always be capitalized.

Thanks for your suggestions, we’ve capitalized all the “Western” in the manuscript where applicable (line 21, line 192 and line 354).

7. IC50, '50' should be subscript.

Thanks for your comments, we’ve subscripted the 50 in the manuscript as well as the figures where IC_50_ appears (line 93, line 110 and line 603-604).

8. Page 6, line 115, the sentence should be fixed.

Thanks for your suggestions and we’ve reconstructed those descriptive sentences to make our expressions clearer, line (113-116).

9. Page 7, line 151, "shifted"? should be "relocated".

Thanks for your suggestion and we’ve changed this word into “relocated” (line 151).

10. Page 7, line 161, the sentence should be fixed.

Thanks for your suggestion and we’ve fixed this sentence to make the manuscript more readable (line 163-164).

11. Page 11, line 241, the sentence should be fixed.

Thanks for your suggestion and we’ve fixed this sentence to make our expression clearer (line 244-246).

12. Page 12, line 264, the sentence should be fixed.

Thanks for your suggestion and we’ve rearranged this sentence (line 269-272).

13. Page 12, line 274, 'displayed' should be 'identified'.

Thanks for your suggestion and we’ve switched this word into “identified” (line 276).

14. Page 13, line 290, 'determine' should be 'assess'.

Thanks for your suggestion and we’ve changed the descriptions in this method part according to other reviewers’ suggestions and we’ve seriously checked our expressions to make sure they are accurate.

15. Figure 1a, the captions of the vertical axis were missing for two panels.

Thanks for your suggestion and we’ve added up the missing vertical axis for the two panels.

16. Figure 1b and 1c, 'Cell Viability' should be 'Cell viability'.

Thanks for your suggestion and we’ve changed the captions in the Figure 1b and 1c into “Cell viability”.

17. Figure 2, 'cell ratio' should be 'Cell ratio'.

Thanks for your suggestion and we’ve changed the captions in the Figure 2 into “Cell ratio”.

18. Figure 4, the font size of the captions should be adjusted to the same.

Thanks for your suggestion. We do understand that keeping the font size of the captions is important so that the figure could be neat and order. However, due to the length of some of the figure captions and the space of the whole figure, we failed to adjust all the captions to the same, otherwise some of the captions will be too small to read. We do adjust most of the font size of the figure captions the same, the bigger size was set to font size 8 and the smaller size was set to font size 6 in Adobe Illustrator. Please find our adjustments in the new figures.

19. Figure 1, Supplement 1, b, the last panel should be adjusted to a circle. The vertical axis, 'colonies formated' should be 'Colony formation'.

Thanks for your suggestion, the panel in Figure 1—figure supplement 1 b have been adjusted to a circle and the figure captions have been adjusted to “Colony formation”.

20. Figure 2, Supplement 1, c, the vertical axis, 'cell ratio' should be 'Cell ratio'. These figure captions should be kept consistent in the paper.

Thanks for your suggestion, the figure captions have been adjusted to “Cell ratio”. We’ve checked other figure captions and kept them consistent in the paper as well as the figures.

Reviewer #2 (Recommendations for the authors):The authors performed preclinical studies to investigate the underlying mechanism of how the combination of pyrotinib, letrozole and dalpiciclib achieved satisfactory clinical outcomes in the MUKDEN 01 clinical trial (NCT04486911). Mechanistically, using anti-HER2 drugs such as pyrotinib and trastuzumab could degrade HER2 and facilitate the nuclear transportation of ER in HER2+HR+ breast cancer, which enhanced the function of ER signaling pathway. The introduction of dalpiciclib partially abrogated the nuclear transportation of ER and exerted its canonical function as cell cycle blockers, which led to the optimal cytotoxicity effect in treating HER2+HR+ breast cancer. Furthermore, using mRNA-seq analysis and in vivo drug susceptibility test, the authors succeeded in identifying CALML5 as a novel risk factor in the treatment of HER2+HR+ breast cancer.

Thanks for your comments and valuable suggestions, we’ve improved our manuscript according to your suggestions below.

1. The catalogue number of the antibodies used in this study shall be added in the section of Chemicals and antibodies.

Thanks for your suggestions, we’ve added the catalogue number of the antibodies used in this study in the “Chemicals and antibodies” part. The information about the antibodies we used could also be found in the Key Resources Table.

2. I suggest the change of "CDK4/6 inhibitor" into "dalpiciclib" in line 221, because only dalpiciclib was used in this study and we were unaware of if other CDK4/6 inhibitors could affect the nuclear transportation of ER.

Thanks for your suggestions, we’ve changed “CDK4/6 inhibitor” into “dalpiciclib” in line 224 to avoid confusion.

Reviewer #3 (Recommendations for the authors):In this research, the authors explore a novel mechanism of CDK4/6 inhibitor dalpiciclib in HER2+HR+ breast cancers, in which dalpiciclib could reverse the process of ER intra-nuclear transportation upon HER2 degradation. The conclusions are significant to gain insight into the biological behavior of TPBC and provided a conceptual basis for the ideal efficacy in the published clinical trial. The findings are supported by supplemented in vivo assay and transcriptomic analysis.

Thanks for your comments and valuable suggestions to us so that we could improve this manuscript.

1. In some parts of the manuscript, the author interchanged the expression of "breast cancer" and "breast tumor", I suggest sticking to one expression throughout the whole text, which may improve the concordance of the manuscript.

Thanks for your suggestions. We realized that interchanging the expression of “breast cancer” and “breast tumor” had caused confusion. We’ve used the expression of “breast cancer” in the whole manuscript (line 43, line 83 and line 158).

2. In Table 1 and Table 2, the reasons for dividing the demographic data into subgroups should be specified in the Material and Method session.

Actually, Table 1 and Table 2 contained patients from the same clinical trial (patients who received chemotherapy+trastuzumab as neoadjuvant therapy and patients who received pyrotinib+dalpiciclib+letrozole as neoadjuvant therapy). In Table 1, patients who only received chemotherapy was not evaluated for the expression of CALML5 before receiving neoadjuvant therapy. To avoid confusions when mentioning the patients who were evaluated for the expression of CALML5 in the manuscript, we additionally made Table 2 to display the demographic characteristics of these patients. We have specified the reasons for dividing the demographic data into subgroups in the Clinical specimen part in the Material and Method session (line 290-294).

3. In Table 1 and Table 2, the first line where different treatment groups were listed should be adjusted for new line management.

Thanks for your suggestion and we’ve adjusted the new line management to make the tables clear and neat.